# GCN meets GPU:
# Decoupling "When to Sample" from "How to Sample"

**Morteza Ramezani** *
Pennsylvania State University
morteza@cse.psu.edu

**Weilin Cong** *
Pennsylvania State University
wxc272@psu.edu

**Mehrdad Mahdavi**
Pennsylvania State University
mzm616@psu.edu

**Anand Sivasubramaniam**
Pennsylvania State University
anand@cse.psu.edu

**Mahmut T. Kandemir**
Pennsylvania State University
kandemir@cse.psu.edu

## Abstract

Sampling-based methods promise scalability improvements when paired with stochastic gradient descent in training Graph Convolutional Networks (GCNs). While effective in alleviating the neighborhood explosion, due to bandwidth and memory bottlenecks, these methods lead to computational overheads in preprocessing and loading new samples in heterogeneous systems, which significantly deteriorate the sampling performance. By decoupling the frequency of sampling from the sampling strategy, we propose LazyGCN, a general yet effective framework that can be integrated with any sampling strategy to substantially improve the training time. The basic idea behind LazyGCN is to perform sampling periodically and effectively recycle the sampled nodes to mitigate data preparation overhead. We theoretically analyze the proposed algorithm and show that under a mild condition on the recycling size, by reducing the variance of inner layers, we are able to obtain the same convergence rate as the underlying sampling method. We also give corroborating empirical evidence on large real-world graphs, demonstrating that the proposed schema can significantly reduce the number of sampling steps and yield superior speedup without compromising the accuracy.

## 1 Introduction

Graphs are powerful and versatile data structures to model many real world problems. However, learning on a graph is challenging because it requires modeling of both rich node features and underlying structure information. In recent years, thanks to its effective representation power and improvements in hardware computing performance, Graph Convolutional Networks (GCNs) [15] and their subsequent variants [13, 25] have achieved great success in numerous domains, including social relationship detection [26, 7, 23], recommender systems [1, 29], knowledge graphs [27, 28, 21], and biological networks [8, 9].

Recent success of training deep neural networks with GPUs, makes such a highly parallel architecture a natural choice for training GCNs. However, directly adopting GPUs on large graphs remains challenging due to computational overheads introduced by inter-dependency between nodes. In fact, different from other standard neural architectures [17, 19, 24, 14] (e.g., fully connected or convolutional neural networks) where the prediction of an individual data sample solely depends on its own features, in a multi-hop GCN, the representation of a node recursively depends on its neighbors across multiple hops (i.e., layers) that need to be aggregated – a phenomenon known

---

as *neighborhood explosion*. Processing this dependency requires node's features, a large portion of estimated features of its neighbors at different hops, along with graph structure to be present in memory, which impedes the scalability to large graphs. This situation is further exacerbated on GPUs where local memory is in general more scarce compared to CPUs. For instance, the memory capacity on a very recent GPU card, such as `NVIDIA Tesla V100`, is at most 32 GB, while a scale-free graph with 50 million nodes can take up to 350 GB.

One way of alleviating this memory demand is to employ *sampling* – an effective strategy that practitioners often use in training GCNs. The aim of sampling-based training is to aggregate the hidden features of only a sampled subset of neighbors at each layer. A number of recent studies have introduced and evaluated different sampling methods such as nodewise sampling [13, 29], layerwise sampling [3, 32], and subgraph sampling [30, 4]. In practice, sampling from a large graph requires many random accesses to the memory, which inherently do not perform well in GPUs, which are designed for regular parallel accesses. A heterogeneous system comprising CPUs and GPUs allows some trade-offs between these two – CPUs are more capable of performing random memory access compared to GPUs, but do not have the high degree of parallelism offered by the latter [31, 18, 20]. However, transferring large volumes of data between the two (CPU and GPU) can further deteriorate the performance.

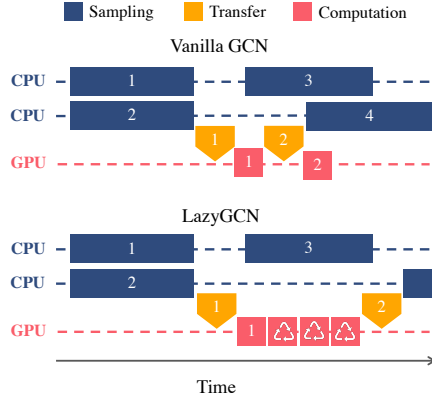

Figure 1: Timeline of executing vanilla GCN vs proposed LAZYGCN on a CPU-GPU system with two processes. The numbered boxes indicate the time spent for each mini-batch at different stages.

For example, as shown in the top half of Figure 1, in training a mini-batch GCN on a heterogeneous system, the majority of the time is spent on sampling nodes using CPU and transferring the sampled nodes from CPU to GPU, rather than the actual computation itself. One viable option to reduce the sampling overhead is to assign more CPU resources; however, this adds computation overhead on CPU and storing the intermediate results of a large number of samplers can cause memory contention. It is worth mentioning that sampling time increases significantly as the graph size grows, while the transfer and computation time on GPUs remain the same, given the limited GPU capacity. Also, a solution to reduce the transfer time is to use smaller mini-batches and leave enough free space on GPU memory for next mini-batch data (i.e. overlap data transfer of next batch with compute of previous batch). However, a smaller mini-batch size in GCN is not preferable as it leads to under utilization of GPU and may also make the algorithm diverge if selected aggressively small.

Motivated by these observations, the key question we investigate in this paper is: *for a given sampling strategy, can we reduce the sampling frequency to leverage the underlying hardware capabilities without compromising accuracy?* In this work, we develop LAZYGCN, a general yet efficient framework that is suitable for heterogeneous settings to train large scale GCNs. As shown in the bottom half of Figure 1, unlike vanilla GCN training methods, where a new sample for each iteration is prepared on CPU and transferred to GPU, LAZYGCN performs periodic sampling on CPU and transfers it to GPU. Then, instead of sampling new data points on CPU at every iteration, for a predetermined number of iterations, LAZYGCN effectively recycles the already-sampled nodes on the GPU, reducing both sampling and data transfer overheads. During recycling, a variance reduction schema is employed to reduce the influence of sampling. While previously proposed techniques such as mini-batch persistency [11] and data echoing [5] aim at incorporating data reuse for training, the main focus of those works are standard deep neural networks, where input dependency is insignificant compared to GCN. In addition, none of these works provide a proper analysis on the convergence of data reuse in the training of deep neural network.

We provide a theoretical analysis that motivates our algorithm, and characterizes its speed of convergence. Indeed, we show that under a mild condition on the recycling size, by reducing the variance of inner layers, we are able to obtain the same convergence rate as the underlying sampling method. We also conduct extensive numerical experiments on different large-scale graph datasets and different sampling methods to corroborate our theoretical findings, and demonstrate the practical efficacy of the proposed algorithm over competitive baselines. Overall, our empirical results demonstrate that LAZYGCN can significantly reduce the number of sampling steps and yield superior speedup without

compromising the accuracy. Finally, we note that, while LAZYGCN is developed for CPU-GPU heterogeneous systems, it is not limited to this particular platform and the proposed techniques can be applied to any heterogeneous or distributed setting where the memory accesses and data transfers form a bottleneck.

**Organization.** The rest of this paper is organized as follows. In Section 2, we review the notations on graph convolutional networks. In Section 3, we propose LAZYGCN, and provide theoretical analysis in Section 4. We empirically evaluate our theory via various experiments in Section 5, and conclude the paper in Section 6. Additional experiments and the proof of convergence rate are deferred to appendix.

## 2   Background

In this paper, we consider the *inductive node classification* problem, where the goal is to predict labels for each node. However, during training, neither the features nor the structure information of the test nodes are present. Although our proposed method can be easily adapted to other settings as well, it has been shown that performing inductive training of GCNs is more challenging [13]. To formally state the problem, suppose $\mathcal{G} = (\mathcal{V}, \mathcal{E})$ is an undirected unweighted graph with $N = |\mathcal{V}|$ nodes and $M = |\mathcal{E}|$ edges, and denote its adjacency and associated degree matrices by $\mathbf{A} \in \{0, 1\}^{N \times N}$ and $\mathbf{D} \in \mathbb{R}^{N \times N}$, respectively. We assume that each node $i \in \mathcal{V}$ is associated with a pair $(\boldsymbol{x}_i, \boldsymbol{y}_i)$, where $\boldsymbol{x}_i \in \mathbb{R}^d$ is the feature vector and $\boldsymbol{y}_i \in \mathbb{R}^C$ is the label vector, e.g., a $C$ dimensional vector in the multi-class classification problem. We use $\mathbf{X} \in \mathbb{R}^{N \times d}$ and $\mathbf{Y} \in \mathbb{R}^{N \times C}$ to denote the feature and label matrix when all nodes are considered. The aim of GCNs is to learn a representation for nodes based on training data $\{(\boldsymbol{x}_i, \boldsymbol{y}_i)\}_{i=1}^N$, while respecting the topology information captured in $\mathcal{G}$, to predict the label on *unseen nodes*. Although the topology is a rich information source that can be leveraged to improve predictive performance, it introduces new challenges in training GCNs. In what follows, we briefly discuss full-batch and sampling-based methods for training GCNs.

**Full-batch training.** Denote symmetric normalized Laplacian matrix as $\mathbf{L} = \mathbf{D}^{-\frac{1}{2}} \mathbf{A} \mathbf{D}^{-\frac{1}{2}}$, and $d_\ell$ as the dimension of hidden features at $\ell$th layer where $d_0 = d$ and $d_L = C$. The $\ell$th layer of full-batch GCN [15] is defined as

$$\mathbf{Z}^{(\ell)} = \mathbf{L} \mathbf{H}^{(\ell-1)} \mathbf{W}^{(\ell)}, \ \mathbf{H}^{(\ell)} = \sigma(\mathbf{Z}^{(\ell)}) \qquad \text{and} \qquad \mathcal{L}(\boldsymbol{\theta}) = \frac{1}{N} \sum_{i \in \mathcal{V}} \phi(\boldsymbol{z}_i^{(L)}, y_i), \qquad (1)$$

where $\sigma(\cdot)$ is the activation function (e.g., ReLU), $\mathbf{W}^{(\ell)} \in \mathbb{R}^{d_{\ell-1} \times d_\ell}$ is the weight matrix at the $\ell$th layer, and $\mathbf{H}^{(\ell)} \in \mathbb{R}^{N \times d_\ell}$ is the feature matrix for all $N$ nodes at the $\ell$th layer with $\mathbf{H}^{(0)} = \mathbf{X}$. The $\boldsymbol{z}_i^{(L)} \in \mathbb{R}^C$ is the final prediction for the node $i$ and $\phi(\cdot)$ is the loss function (e.g., cross-entropy). Our goal is to learn a set of stacked parameters $\boldsymbol{\theta} = \{\mathbf{W}^{(1)}, \dots, \mathbf{W}^{(L)}\}$ by minimizing the empirical loss $\mathcal{L}(\boldsymbol{\theta})$ over all nodes in the training set.

**Sampling-based training.** On a large graph, training full-batch GCN is computation prohibitive and requires a very large memory footprint. To overcome this issue, **mini-batch GCN** training methods are proposed where, instead of using all training nodes $\mathcal{V}$ for each iteration, a mini-batch of nodes $\mathcal{V}_\mathcal{B}$ of size $B$ are selected from $\mathcal{V}$ and used to calculate the empirical loss as $\mathcal{L}_\mathcal{B}(\boldsymbol{\theta}) = \frac{1}{B} \sum_{i \in \mathcal{V}_\mathcal{B}} \phi(\boldsymbol{z}_i^{(L)}, \boldsymbol{y}_i)$.

Although mini-batch GCN training method can alleviate the computation issue, due to *neighbor explosion*, node dependency for a single batch can quickly cover the entire graph as the number of layers increases. To further address this issue, sampling-based training methods [13, 32, 3, 30, 4, 6, 2] (including nodewise, layerwise and subgraph sampling) are proposed to sample subset of nodes in each layer to construct a smaller Laplacian matrix $\widetilde{\mathbf{L}}^{(\ell)}$ from $\mathbf{L}$. Let us consider nodewise sampling from GraphSAGE [13] as an example. For each node in layer $\ell$, a fixed number of neighbor nodes are sampled to build the sampled Laplacian matrix $\widetilde{\mathbf{L}}^{(\ell)}$. Hence, the graph convolution and empirical loss can be computed as:

$$\widetilde{\mathbf{Z}}^{(\ell)} = \widetilde{\mathbf{L}}^{(\ell)} \widetilde{\mathbf{H}}^{(\ell-1)} \mathbf{W}^{(\ell)}, \ \widetilde{\mathbf{H}}^{(\ell)} = \sigma(\widetilde{\mathbf{Z}}^{(\ell)}) \quad \text{and} \quad \widetilde{\mathcal{L}}_\mathcal{B}(\boldsymbol{\theta}) = \frac{1}{B} \sum_{i \in \mathcal{V}_\mathcal{B}} \phi(\widetilde{\boldsymbol{z}}_i^{(L)}, y_i), \qquad (2)$$

where $\widetilde{\boldsymbol{z}}_i^{(L)} \in \mathbb{R}^C$ as the feature for the node $i$ in $\widetilde{\mathbf{Z}}^{(L)}$. Unlike nodewise, in layerwise sampling each layer has fixed (and smaller) number of sampled nodes, shared between all nodes in that layer. Subgraph sampling is similar to layerwise sampling by restricting $\widetilde{\mathbf{L}}^{(1)} = \cdots = \widetilde{\mathbf{L}}^{(L)}$. It is worth

**Algorithm 1** LAZYGCN training algorithm

---

**Input:** Number of layers $L$, node features $\mathbf{X} \in \mathbb{R}^{N \times d}$, labels $\mathbf{Y} \in \mathbb{R}^{N \times C}$, recycling period size $R$, recycling growth rate $\rho$, learning rate $\eta$, and sampling strategy SAMPLER

  1: **for** $k \leftarrow 1$ **to** $K$ **do**

  2:      Sample a *recycling* mini-batch $\mathcal{V}_k^{(L)}$ of size $S$                ▷ Taking fresh samples

  3:      Sample inner layer nodes $\mathcal{V}_k^{(L-1)}, \ldots, \mathcal{V}_k^{(0)}$ for nodes in $\mathcal{V}_k^{(L)}$ using SAMPLER

  4:      Construct $\widetilde{\mathbf{L}}^{(1)}, \ldots, \widetilde{\mathbf{L}}^{(L)}$ where $\widetilde{\mathbf{L}}^{(\ell)} \in \mathbb{R}^{|\mathcal{V}_k^{(\ell)}| \times |\mathcal{V}_k^{(\ell-1)}|}$

  5:      Transfer node features $\{\mathbf{x}_v | v \in \mathcal{V}_k^{(0)}\}$ and $\widetilde{\mathbf{L}}^{(1)}, \ldots, \widetilde{\mathbf{L}}^{(L)}$ to GPU

  6:      $\boldsymbol{\theta}_{k,1} \leftarrow \boldsymbol{\theta}_k$

  7:      **for** $r \leftarrow 1$ **to** $\rho^k R$ **do**                                     ▷ Recycling

  8:          **for** $\ell \leftarrow 1$ **to** $L$ **do**

  9:              Estimate embedding matrices

$$\widetilde{\mathbf{Z}}^{(\ell)} = \widetilde{\mathbf{L}}^{(\ell)} \widetilde{\mathbf{H}}^{(\ell-1)} \mathbf{W}_{k,r}^{(\ell)} \quad \text{and} \quad \widetilde{\mathbf{H}}^{(\ell)} = \sigma\left(\widetilde{\mathbf{Z}}^{(\ell)}\right)$$

10:          **end for**

11:      Sample a subset of mini-batch $\mathcal{V}_{k,r} \subseteq \mathcal{V}_k^{(L)}$ of size $B$

12:      Compute gradient for nodes in $\mathcal{V}_{k,r}$ as

$$\mathbf{g}_{k,r} = \frac{1}{B} \sum_{i \in \mathcal{V}_{k,r}} \nabla \phi\left(\widetilde{\boldsymbol{z}}_i^{(L)}, \boldsymbol{y}_i\right), \quad \text{where } \widetilde{\mathbf{Z}}^{(L)} = [\widetilde{\boldsymbol{z}}_1^{(L)}, \ldots, \widetilde{\boldsymbol{z}}_N^{(L)}]$$

13:          Update model parameters by $\boldsymbol{\theta}_{k,r+1} = \boldsymbol{\theta}_{k,r} - \eta \mathbf{g}_{k,r}$

14:      **end for**

15:      $\boldsymbol{\theta}_{k+1} \leftarrow \boldsymbol{\theta}_{k,r+1}$

16: **end for**

**Output:** GCN model with trained weight $\boldsymbol{\theta}_K = \{\mathbf{W}_K^{(1)}, \ldots, \mathbf{W}_K^{(L)}\}$

---

noting that the sampled Laplacian matrix leads to a much lower computational complexity and memory requirement, makes it possible to use current available systems.

## 3 Proposed algorithm

In this section, we introduce our LAZYGCN algorithm that incorporates two key modules, (i) periodic sampling with decreasing frequency and (ii) effective recycling with variance reduction, to mitigate sampling and data transfer overheads without compromising the predictive accuracy. Indeed, a crucial question that we need to address is finding the optimal size of recycling period for minimizing the overall sampling cost of the training. Choosing a large value of recycling period reduces the number of sampling steps for a fixed number of iterations, which leads to a significant reduction in data preparation time. However, aggressively increasing the recycling period is not necessarily optimal, as the algorithm overfits to the sampled nodes in the mini-batch. The proposed method is formally summarized in Algorithm 1. We describe the details of its steps below.

**(1) Periodic sampling.** Let us define $T$ as the total number of iterations to update the model parameters. In the vanilla sampling-based GCN, at each iteration, we randomly sample a fresh mini-batch of nodes and their corresponding inner layer neighbors to update the model parameters via recursive aggregation, which suffers from an $O(T)$ data preparation overhead. To reduce the frequency of sampling, the proposed method consists of $K \ll T$ epochs, where at the beginning of the $k$th epoch, we sample a fresh mini-batch of nodes $\mathcal{V}_k^{(L)}$ of size $S$ for recycling (line 2). Next, the inner-layers node are selected using a GCN sampler (SAMPLER), that could be any of the previously-mentioned strategies, and the sampled node features $\{\mathbf{x}_v | v \in \mathcal{V}_k^{(0)}\}$ and Laplacians $\widetilde{\mathbf{L}}^{(1)}, \ldots, \widetilde{\mathbf{L}}^{(L)}$, are built and transferred to the GPU (line 3-5).

The $k$th recycling epoch proceeds for $\rho^k R$ iterations, where $R$ is the base *recycle period size* and $\rho \geq 1$ is the *recycling growth rate* (lines 7-14). This results in a total of $T = \sum_{k=1}^{K} \rho^k R$ iterations, which require $K = \log_\rho^{T/R}$ fresh samplings. We note that, for a fixed number of recycling steps, i,e., $\rho = 1$, we only need to sample fresh nodes $K = \frac{T}{R}$ times. Although this could lead to reduction in

the data preparation overhead, we instead propose to employ progressively larger recycling periods by picking $\rho > 1$, which significantly saves the computational overhead related to preparing new mini-batches. The main intuition behind this idea stems from the following observation about the dynamic of optimization algorithm: *a smaller recycling period size seems advantageous early on, as the model parameters are far from the optimal solution, and more recycling on a sampled mini-batch might cause an overfit. However, as the algorithm approaches an optimal solution where the gradient vanishes, it seems more reasonable to perform more recycling steps.*

**(2) Recycling with variance reduction.** During the recycling stage, we perform a vanilla sampling-based GCN over the sampled nodes $\mathcal{V}_k^{(L)}$, with a key difference that we *fix* the inner layer nodes. As explained before, in the standard training of GCNs, the inner layer sampling is utilized to overcome the neighbor explosion issue and scale well with the number of nodes (this is essential for applying GCN to large graphs). On the downside, the variance of gradient for the sampled mini-batch suffers from an additional bias due to the sampling of inner layers as we approximate the inner layers in the forward pass to compute the stochastic gradients over the sampled mini-batch. This suggests that a variance reduction schema in forward pass can be put into place to lessen the effect of sampling and speed up the convergence. Intuitively speaking, concentrating the optimization on fewer nodes in $\mathcal{V}_{k,r}$ with more accurate gradient estimations (thanks to variance reduction in forward pass) may be better than updating over all nodes $\mathcal{V}_k^{(L)}$ with highly noisy gradients.

Motivated by this observation, we fix the inner layer nodes while recycling (lines 8-10), to those sampled at the beginning of the recycling stage $\mathcal{V}_k^{(\ell)}, \ell = 0, ..., L-1$ (line 3), to explicitly reduce the variance. Then, at each iteration of the recycling, we sample $B$ nodes ($\mathcal{V}_{k,r}$) uniformly at random from the last layer nodes $\mathcal{V}_k^{(L)}$ to update the model using their gradients (lines 11-13). In this way, we can consider the recycling as an application of vanilla SGD over the samples in $\mathcal{V}_k^{(L)}$, where the variance of the forward pass is fixed to one induced by samples in $\mathcal{V}_k^{(1)}, \ldots, \mathcal{V}_k^{(L-1)}$. As we elaborate later in Lemma 2, by reducing the variance of inner layers, the effects of the sampling become negligible and, as a result, we can obtain the same convergence rate as the underlying sampling method.

## 4   Theoretical analysis

In this section, we establish the convergence rate of LazyGCN algorithm and show that, under mild assumptions, LazyGCN enjoys the same convergence rate as underlying sampling algorithm employed in training. Specifically, we show that periodic sampling with implicit variance reduction (i.e., fixing the inner layers during recycling) convergences as fast as fresh sampling over all layers at each iteration. While our results generalize to an arbitrary number of layers, for the ease of exposition, we focus on a single-layer GCN to illustrate the main ideas and key techniques. Note that $L$-layer GCN can be formulated as $L + 1$ level optimization problem where all challenges in $L$-layer GCNs (e.g., biased stochastic gradient and the effect of variance at the $\ell$th layer on variance at the $\ell + 1$th layer) also exist in the single layer GCN. We denote the loss functions of full-batch, mini-batch and sampling-based GCN as

$$F(\boldsymbol{\theta}) = \frac{1}{N} \sum_{i \in \mathcal{V}} f_i\Big( \frac{1}{|\mathcal{N}(i)|} \sum_{j \in \mathcal{N}(i)} g_j(\boldsymbol{\theta}) \Big), \qquad \text{(full-batch)}$$

$$F_{\mathcal{B}}(\boldsymbol{\theta}) = \frac{1}{B} \sum_{i \in \mathcal{V}_{\mathcal{B}}} f_i\Big( \frac{1}{|\mathcal{N}(i)|} \sum_{j \in \mathcal{N}(i)} g_j(\boldsymbol{\theta}) \Big), \qquad \text{(mini-batch)}$$

$$\widetilde{F}_{\mathcal{B}}(\boldsymbol{\theta}) = \frac{1}{B} \sum_{i \in \mathcal{V}_{\mathcal{B}}} f_i\Big( \frac{1}{|\widetilde{\mathcal{N}}(i)|} \sum_{j \in \widetilde{\mathcal{N}}(i)} g_j(\boldsymbol{\theta}) \Big), \qquad \text{(sampling-based)}$$

respectively, where the outer and inner layer function are defined as $f(\cdot) \in \mathbb{R}$ and $g(\cdot) \in \mathbb{R}^d$, and their gradients as $\nabla f(\cdot) \in \mathbb{R}^d$ and $\nabla g(\cdot) \in \mathbb{R}^{d \times d}$, respectively. Notice that, compared to vanilla SGD training, the key challenge in the sampling-based GCN is due to its biased stochastic gradient as the gradient becomes biased because of inner layer sampling, i.e., $\mathbb{E}[\nabla \widetilde{F}_{\mathcal{B}}(\boldsymbol{\theta})] \neq \nabla F(\boldsymbol{\theta})$. For theoretical analysis, we suppose every node in the graph has the same number of neighbors and make the following standard assumptions on the smoothness of $f(\cdot), g(\cdot)$ and their gradients $\nabla f(\cdot), \nabla g(\cdot)$.

**Assumption 1.** *Suppose $f(\cdot)$ is $L_f$-Lipschitz continuous, $g(\cdot)$ is $L_g$-Lipschitz continuous, $\nabla f(\cdot)$ is $G_f$-Lipschitz continuous, $\nabla g(\cdot)$ is $G_g$-Lipschitz continuous.*

A key quantity that affect the convergence of stochastic optimization algorithm is the *mean-square error (MSE)* of stochastic gradient to the full gradient. As shown in Eq. 3, the MSE can be decomposed into representation approximation error $\mathrm{Err}_1(t)$ and gradient approximation error $\mathrm{Err}_2(t)$.

$$\frac{1}{2}\mathbb{E}\left[\left\|\nabla\widetilde{F}_{\mathcal{B}}(\boldsymbol{\theta}_t) - \nabla F(\boldsymbol{\theta}_t)\right\|^2\right] \leq \underbrace{\mathbb{E}\left[\left\|\nabla\widetilde{F}_{\mathcal{B}}(\boldsymbol{\theta}_t) - \nabla F_{\mathcal{B}}(\boldsymbol{\theta}_t)\right\|^2\right]}_{\mathrm{Err}_1(t)} + \underbrace{\mathbb{E}\left[\left\|\nabla F_{\mathcal{B}}(\boldsymbol{\theta}_t) - \nabla F(\boldsymbol{\theta}_t)\right\|^2\right]}_{\mathrm{Err}_2(t)}. \quad (3)$$

The above decomposition is crucial in the convergence rate and will be utilized in our analysis. To be able to compare the convergence of LAZYGCN to its underlying sampling algorithm (without lazy sampling), we first state the convergence of sampling-based GCN (SGCN) in the following theorem.

**Theorem 1.** *Suppose Assumption 1 holds and let $L_F = L_f G_g + L_g^2 G_f$ and*

$$\Delta_{SGCN} = \mathcal{O}\left(L_f^2 G_g^2 \frac{\log(2d/\delta)}{\widetilde{D}}\right) + \mathcal{O}\left(G_f^2 L_g^4 \frac{\log(2d/\delta) + 1/4}{\widetilde{D}}\right) + \mathcal{O}\left(G_f^2 \frac{\log(2d/\delta) + 1/4}{B}\right),$$

*where $B$ and $\widetilde{D}$ are the size and the number of neighbors in a random mini-batch. By choosing $\eta = \min\left\{\frac{3}{2L_F}, \sqrt{\frac{\mathbb{E}[F(\boldsymbol{\theta}_1)] - \mathbb{E}[F(\boldsymbol{\theta}_\star)]}{\Delta_{SGCN} L_F T}}\right\}$, for $\tilde{\boldsymbol{\theta}} = \min_t \mathbb{E}\left[\|\nabla F(\boldsymbol{\theta}_t)\|\right]$, with probability at least $1 - \delta$, it holds that*

$$\mathbb{E}[\|\nabla F(\tilde{\boldsymbol{\theta}})\|^2] \leq \mathcal{O}(\sqrt{\Delta_{SGCN}/T}). \quad (4)$$

The above theorem implies that to find a solution such that $\mathbb{E}[\|\nabla F(\tilde{\boldsymbol{\theta}})\|^2] \leq \epsilon$ holds, the SGCN requires to iterate over $T = \Delta_{\mathrm{SGCN}}/\epsilon^2$ fresh mini-batches.

### 4.1 Convergence analysis for LAZYGCN

We now turn to characterizing the convergence of LAZYGCN. First, in the following lemma we show that fixing the inner layer sampling inside each recycling iteration will *not* hurt the representation approximation error $\mathrm{Err}_1(t)$.

**Lemma 1.** *With probability at least $1 - \delta$ we have*

$$\frac{1}{2}\mathbb{E}\left[\left\|\nabla\widetilde{F}_{\mathcal{B}}(\boldsymbol{\theta}) - \nabla F_{\mathcal{B}}(\boldsymbol{\theta})\right\|^2\right] \leq 16 L_f^2 G_g^2 \frac{\log(2d/\delta)}{\widetilde{D}} + 32 G_f^2 L_g^4 \frac{\log(2d/\delta) + 1/4}{\widetilde{D}}, \quad (5)$$

*where $\widetilde{D}$ is the number of selected neighbors in the sampling.*

In the next lemma, we show that sampling a mini-batch of $B$ nodes from a subset of nodes instead of all nodes, will *not* result in a large gradient approximation error $\mathrm{Err}_2(t)$.

**Lemma 2.** *Consider the setting where we sample a mini-bath of size $B$ from a subset of $S$ nodes uniformly at random instead of sampling from $N$ nodes. Then, with probability at least $1 - \delta$ we have*

$$\frac{1}{2}\mathbb{E}\left[\|\nabla F_{\mathcal{B}}(\boldsymbol{\theta}) - \nabla F(\boldsymbol{\theta})\|^2\right] \leq 16 G_f^2 \frac{\log(2d/\delta) + 1/4}{B} + 16 G_f^2 \frac{\log(2d/\delta) + 1/4}{S}, \quad (6)$$

*provided that $S \geq \frac{G_f^2}{L_f^2 G_g^2 + 2 G_f^2 L_g^4} \widetilde{D}$.*

**Proposition 1.** *By the error decomposition in Eq. 3, Lemma 1, and Lemma 2, with probability at least $1 - \delta$, it holds that*

$$\begin{aligned}
\Delta_{\mathrm{LAZYGCN}} &= \mathcal{O}\left(L_f^2 G_g^2 \frac{\log(2d/\delta)}{\widetilde{D}}\right) + \mathcal{O}\left(G_f^2 L_g^4 \frac{\log(2d/\delta) + \frac{1}{4}}{\widetilde{D}}\right) \\
&\quad + \mathcal{O}\left(G_f^2 \frac{\log(2d/\delta) + \frac{1}{4}}{B}\right) + \mathcal{O}\left(G_f^2 \frac{\log(2d/\delta) + \frac{1}{4}}{S}\right).
\end{aligned} \quad (7)$$

Equipped with above results, we provide the convergence of LAZYGCN in the following theorem.

**Theorem 2.** *By setting step size as* $\eta = \min\left\{\frac{3}{2L_F}, \sqrt{\frac{\mathbb{E}[F(\boldsymbol{\theta}_1)] - \mathbb{E}[F(\boldsymbol{\theta}_\star)]}{L_F(\sum_{k=1}^K \rho^k R\Delta_{\text{LAZYGCN}})}}\right\}$, *for* $\tilde{\boldsymbol{\theta}} = \min_t \mathbb{E}[\|\nabla F(\boldsymbol{\theta}_t)\|]$, *we have*

$$\mathbb{E}[\|\nabla F(\tilde{\boldsymbol{\theta}})\|^2] \leq \mathcal{O}\left(\frac{\sqrt{\sum_{k=1}^K \rho^k R\Delta_{\text{LAZYGCN}}}}{T}\right). \tag{8}$$

To compare the above bound to the one obtained for SGCN in Theorem 1, consider the case where $\rho = 1$. In this case the rate of LAZYGCN reduces to $\sqrt{\Delta_{\text{LAZYGCN}}/T}$, same as the convergence rate of SGCN, but with a difference in the MSE of stochastic gradients, i.e., $\Delta_{\text{LAZYGCN}}$ versus $\Delta_{\text{SGCN}}$. As a result, the residual error incurred by sampling a mini-batch from a subset of nodes with size $S$ during recycling stage instead of fresh sampling from all nodes, is the key distinguishing factor between the two rates, which can be bounded by $\Delta_{\text{LAZYGCN}} - \Delta_{\text{SGCN}} = \mathcal{O}\left(G_f^2 \frac{\log(2d/\delta)+1/4}{S}\right)$. As we can see, the scale of the residual error is inversely proportional to the size of recycling samples, and the residual error approaches zero when $S = |\mathcal{V}|$.

## 5 Experimental evaluation

**Datasets and setup.** We evaluate the effectiveness of LAZYGCN under inductive supervise setting on the following real-world datasets: `Pubmed`, `PPI-Large`, `Flickr`, `Reddit`, `Yelp`, and `Amazon`. Detailed information of these datasets are summarized in Table 1. Although the LazyGCN is more effective on large graphs, we used several available datasets from recent related works to evaluate the accuracy of using LAZYGCN.

We applied LAZYGCN on three state-of-the-art sampling techniques, namely, nodewise (Graph-SAGE [13]), layerwise (LADIES [32]) and subgraph (GraphSAINT [30]). We implemented all these algorithms alongside LAZYGCN, using PyTorch [22] and PyTorch Geometric [10] for sparse matrix operations. Further details of our experimental settings are deferred to the appendix. For nodewise, we used 5 neighbors to sample, for layerwise we used a sample size of 512, and for subgraph we used a sample size which is equal to the mini-batch size. For LAZYGCN training, we used fixed $R = 2$ and $\rho = 1.1$ unless otherwise stated. All our experiments are conducted using a 3-layer GCN with hidden dimension of 512 and `Adam` optimizer with a learning rate of $10^{-3}$. Test and validation accuracies (`F1` score) are obtained by running the full-batch GCN.

Table 1: Summary of datasets statistics. ‡ indicates multi-labels dataset

| Dataset | Nodes | Edges | Degree | Feature | Classes | Train / Validation / Test |
|---|---|---|---|---|---|---|
| Pubmed | 19,717 | 44,338 | 3 | 500 | 3 | 92% / 3% / 5% |
| PPI-Large | 56,944 | 1,612,348 | 15 | 50 | 121‡ | 66% / 12% / 22% |
| Flickr | 89,250 | 899,756 | 10 | 500 | 7 | 50% / 25% / 25% |
| Reddit | 232,965 | 11,606,919 | 50 | 602 | 41 | 66% / 10% / 24% |
| Yelp | 716,847 | 13,954,819 | 19 | 300 | 100‡ | 75% / 15% / 10% |
| Amazon | 1,598,960 | 264,339,468 | 124 | 200 | 107 | 78% / 5% / 15% |

### 5.1 Primary results

We evaluate the effect of using LAZYGCN on training accuracy and wall-clock time of three vanilla sampling-based GCNs. For all experiments we executed training five times, and report the mean for test `F1` score and training time in Table 2. It can be observed from this table that, LAZYGCN is capable of reducing the training wall-clock time, regardless of sampling techniques, while maintaining the same degree of accuracy. In smaller datasets, such as `Pubmed` and `Flickr`, the reduction in the training time is negligible, due to the overheads associated with the other parts of the system (e.g., function launches and transfer initiation). However, with larger datasets, such as `Reddit` and `Yelp`, using LAZYGCN brings up significantly better training times, while (in both cases) not impacting accuracy in any significant way.

In the case of `Reddit`, the validation scores per wall-clock and iterations are plotted in Figures 2 (a) and 2 (b), respectively, when using layerwise and subgraph samplings and their LAZYGCN equivalents. These results clearly indicate that LAZYGCN does not just speed up the training time, but also

Table 2: Comparison of the accuracy (F1 score) and training time (seconds) of LAZYGCN and its base methods (including nodewise sampling GraphSAGE, layerwise sampling LADIES, and subgraph sampling GraphSAINT) on GCN training

|  |  | Nodewise | | Layerwise | | Subgraph | |
|---|---|---|---|---|---|---|---|
|  |  | Vanilla | LAZYGCN | Vanilla | LAZYGCN | Vanilla | LAZYGCN |
| Pubmed | F1 | 84.76% | 84.80% | 85.28% | 85.00% | 85.04% | 84.84% |
|  | Time (s) | 10.89 | **3.53** | 0.50 | **0.25** | 0.13 | **0.11** |
| PPI-Large | F1 | 47.89% | 47.94% | 53.81% | 53.31% | 54.52% | 54.43% |
|  | Time (s) | 165.82 | **4.61** | 4.07 | **1.15** | 3.24 | **1.67** |
| Flickr | F1 | 48.63% | 48.61% | 48.72% | 48.74% | 49.02% | 48.75% |
|  | Time (s) | 45.22 | **6.96** | 1.13 | **0.37** | 0.59 | **0.23** |
| Reddit | F1 | 93.26% | 93.37% | 93.49% | 93.38% | 93.91% | 93.96% |
|  | Time (s) | 968.60 | **43.85** | 10.46 | **2.01** | 5.15 | **2.01** |
| Yelp | F1 | 62.78% | 61.96% | 60.07% | 60.11% | 63.24% | 63.28% |
|  | Time (s) | 2169.70 | **174.49** | 227.04 | **77.62** | 106.31 | **48.33** |
| Amazon | F1 | 77.29% | 76.99% | 77.23% | 77.10% | 77.25% | 77.12% |
|  | Time (s) | 5092.8 | **463.9** | 571.4 | **206.8** | 385.2 | **198.4** |

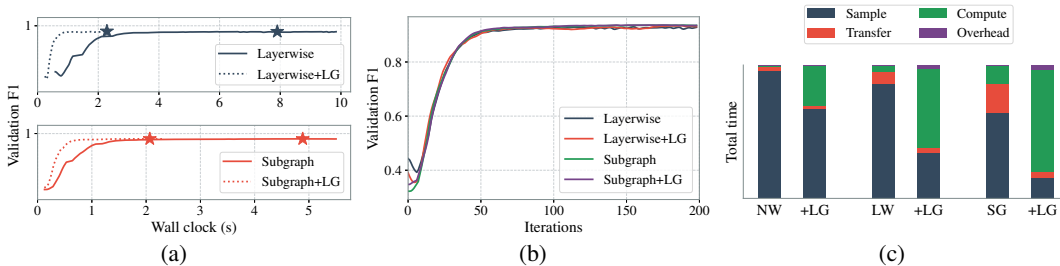

Figure 2: (a) LAZYGCN performance (dashed line) vs. vanilla sampling based GCN (solid line) (b) Training loss per iterations for vanilla and LAZYGCN (c) Breakdown of various GCN training stages. LW stands for layerwise, NW and SG are short for nodewise and subgraph sampling, respectively. Note that the total time is normalized to better show the fraction of time spent in each stage.

achieves the same convergence as the underlying sampling algorithm, further confirming our analysis in Section 4.

**Comparison of the fraction of time on different training stages.** We summarize in Figure 2 (c) the breakdown of the time spent in each stage of GCN training (sampling, transfer, and compute) for three different samplers and their LAZYGCN analogous. Note that the sampling of $\mathcal{V}_{k,r}$ from $\mathcal{V}_k$ in LAZYGCN introduces an additional overhead that is negligible compared to the other stages, as illustrated in this figure. For nodewise, due to per nodes operations, the sampling overhead is extremely large and, as shown in Table 2, this method has much slower execution compared to the other two methods. However, using LAZYGCN, the training time can be reduced by a few orders of magnitude, since LAZYGCN targets data preparation, i.e. *both* sampling and transfer.

## 5.2 Ablation study

In this section we explore the effect of various parameters in LAZYGCN.

**The effect of recycling mini-batch size $S$.** We compare the effect of different mini-batch sizes $S$ on LAZYGCN in Figure 3 (a). As can be observed from this plot, a small mini-batch size has the advantage of less data preparation. However, it also tends to limit the overall performance, and usually requires many more iterations. On the other hand, a very large $S$ slows down the sampling and transfer, and hurts the overall speed. Nevertheless, we observe that, in all the cases tested (except for the very small $S$), LAZYGCN can achieve the same accuracy as the underlying sampling method, with much faster speed compared to vanilla GCN.

**The effect of recycle period size $R$.** Figure 3 (b) compares the effect of different values of $R$ on the convergence speed and accuracy of the models. While it can effect the time, as long as the $R$ is not

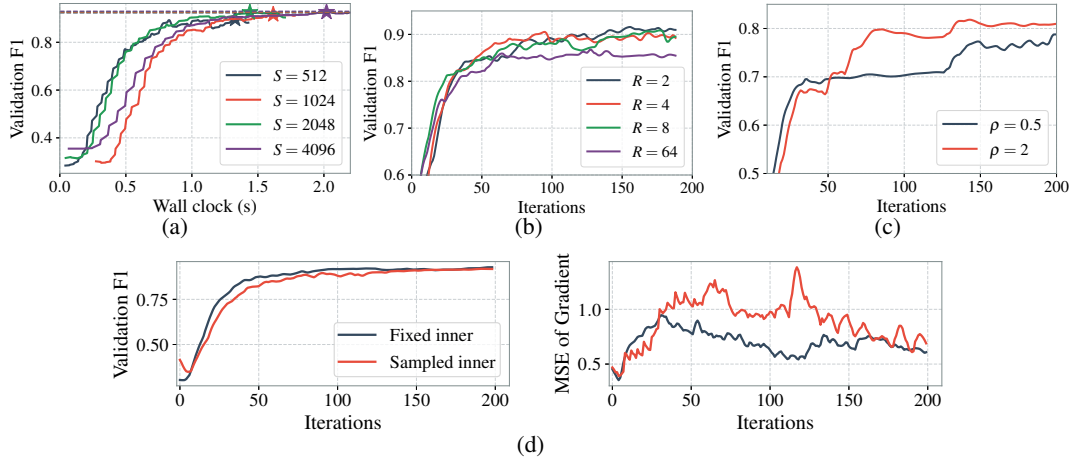

Figure 3: Comparison of validation accuracy on Reddit dataset under different settings (a) Effect of $S$ values. Dashed line shows the vanilla's best score which is $5\times$ slower on average. (b) Different $R$ value (c) Different $\rho$ value (d) Validation and MSE of gradient for fixed inner vs. sampled inner

extremely large and $\mathcal{V}_k^{(L)}$ is chosen carefully, it does not affect the performance. However, when $R$ is very large, it can overfit to the selected recycling mini-batch and requires more iterations.

**The effect of recycling growth rate $\rho$.** We demonstrate the effect of $\rho > 1$ and $\rho < 1$ on the convergence of LazyGCN with layerwise method in Figure 3 (c). For $\rho = 0.5$ we set $R = 128$ and for $\rho = 2$ we set $R = 1$ to have fair comparison between these two settings. It can be seen that with $\rho < 1$, we require more iterations to recover from the overfitted model, however using $\rho > 1$ does not suffer from this issue.

**The effect of fixing inner layer during recycling epoch.** We also conducted an experiment to study the effect of fixing the inner layers in LazyGCN. Figure 3 (d) plots the validation score per iterations and *Mean Square Error* (MSE) for stochastic gradients for two different methods – fixed inner-layer as in LazyGCN and sampled inner-layer. Note that we chose the same inner layer sample size for fair comparison. As can be observed from this figure, LazyGCN, by fixing the inner layer, does not just achieve better variance but also better convergence. It is worth noting that, sampling from inner layer nodes on the GPU can cause significant slow down, due to random access inefficiency which is inherent in GPUs, that makes the second method more inefficient.

## 6  Conclusion

In this paper, we presented LazyGCN, a new method for training Graph Convolutional Networks (GCNs) for large graphs on heterogeneous systems. We discovered that, as opposed to other neural networks, GNNs are not only more input dependent but also data preparation can take longer due to random structure of graphs. To overcome these issues, we introduce a lazy but smart method to reuse the sampled mini-batch more often. Our proposed method brings up significant reductions in training time on available graph dataset. In addition, we provide theoretical analysis to explain why LazyGCN works and why it does not sacrifice accuracy. We believe that LazyGCN also opens the path for learning GCNs in distributed systems where either the single worker or the communication can slow down the overall performance.

## Acknowledgements

This work was supported in part by CRISP, one of six centers in JUMP, a Semiconductor Research Corporation (SRC) program sponsored by DARPA and NSF grants 1909004, 1714389, 1912495, 1629915, 1629129, 1763681, 2008398.

## Broader Impact

The proposed LAZYGCN algorithm has two applicable aspects: system and theory. In terms of system design it is aiming at improving the main bottleneck of GCN training, however its application is not limited to CPU-GPU system, but any other distributed training setting, where either workers or communication is at disadvantage. In addition, the analysis of theoretical aspects of lazy sampling, provides powerful tools for future studies of sampling-based GCNs.

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
