[Supplementary Material]

# Supplementary Material
# GCN meets GPU:
# Decoupling "When to Sample" from "How to Sample"

**Organization.** The appendix is organized as follows. In Section A, we provide the detailed experimental setups and additional results. To reproduce the results reported in this paper, a pointer to the code is provided as well. In Section B, we present a detailed summary of all notations used throughout the paper. We provide the proof of Theorem 1 and Theorem 2, in Section C and Section D, respectively. In Section E, we summarize the useful lemmas and theorems that facilitate our analysis.

## A Detailed experimental setup

### A.1 Hardware specifications and environment

We implement a unified platform to run the GCN models, with PyTorch 1.5 compiled with CUDA 10.2 and PyTorch Geometric [10] for sparse matrix operations. We run our experiments on a machine equipped with Intel Xeon 6320 with 768GB RAM and NVIDIA Quadro RTX 8000 with 48GB memory. While this GPU is capable of running most of the available datasets in full-batch, in our experiments, we assume a limited memory budget, wherein the corresponding mini-batch adjacency structure and node features need to be transferred to the GPU memory every iteration. For each dataset, the mini-batch size is selected to be proportional to the size of the dataset to reflect this limitation. For fair comparison we use the same budget on the sampling as well. To report the time, we only measure the training phases on both CPU and GPU, which includes: sampling, transferring data, and forward and backward pass.

**Implementation code.** The codes for our experiments are available at this repository. Please refer to README.md for the instructions of how to install and run the codes.

### A.2 Dataset details

In Table 3 we summarize the description of each dataset used to evaluate LazyGCN. Also we show the mini-batch size we use for the experiments in Table 2.

Table 3: Datasets description and mini-batch size used for the main experiment

| Dataset | Mini-batch | Description |
|---|---|---|
| Pubmed [7] | 4,096 | Academic papers classified into categories based on citations and content |
| PPI-Large [7] | 8,192 | Protein functions classified based on the interactions of human tissue proteins |
| Flickr [7] | 8,192 | Images classified based on descriptions and common properties of online images |
| Reddit [30] | 16,384 | Online posts from reddit classified into communities based on users comments and links |
| Yelp [30] | 65,536 | Product classified into categories based on customer reviews and relationship |
| Amazon [4] | 65,536 | Product classified into categories based on customer reviews and relationship |

## A.3 Ablation study

In this section we explore the effect of various parameters in LAZYGCN.

**Comparing different values of recycle period size $R$.** Similar to Figure 3 (b), in this experiment, we compare the effect of various recycling period size $R$ on the training loss per iteration for layerwise sampling. In Figure 4, as $R$ increases, the model will over-fit more on the selected recycling mini-batches. However, as previously explained in Section 5, if $R$ is not chosen aggressively large, the models will quickly recover and will not hurt the performance.

(a) Pubmed

(b) Reddit

Figure 4: Comparison of training loss on `Pubmed` and `Reddit` for different recycling period size $R$.

**Comparing various recycling growth rate $\rho$.** Similar to previous experiment, in Figure 5, we evaluate the validation score and training loss for various $\rho$ values on two different datasets. Once again, when $\rho$ is large, the training loss can change drastically as new recycling mini-batch is created. This can lead to lower accuracy as shown in Figure 5 for $\rho = 4$. Note that for `Pubmed` dataset, while it seems that $\rho = 2$ shows less validation accuracy compared to $\rho = 4$, it is not the case, as it reaches much higher validation accuracy in earlier iterations (e.g., 25 iterations).

(a) Pubmed

(b) Reddit

Figure 5: Comparison of training loss and validation accuracy for different $\rho$.

**Effect of fixing inner-layers.** Similar to Figure 3 (d), we compare the validation accuracy and mean-square error of stochastic gradients for different values of $S$. As shown in Figure 6 and Figure 7, fixing the inner layers inside the recycling epoch (e.g., as we do in LAZYGCN) results in a faster convergence rate and a lower variance, when comparing to the case where we sample nodes also in the inner layers. Note that for the sake of fair comparison, we use the same inner layer size for both configurations. We can also observe that as the algorithm reaches to the final solution, the variance shrinks, which matches our intuition of using large recycling steps later in optimization.

Figure 6: Validation accuracy and mean-square error of stochastic gradients on `Reddit` dataset.

Figure 7: Validation accuracy and mean-square error of stochastic gradients on `Pubmed` dataset.

**Comparison of GCN training stages for LAZYGCN.** In Table 4, we report the fraction of time spends in each stage of training in LAZYGCN and the baseline on `Reddit` dataset. LAZYGCN greatly improves the speed of training models and reduces both sampling time and transfer time. We can also see the computation and transfer time become negligible compared to the time spent on sampling as the size of input graphs increases, which is the key motivation for our proposal. This table also shows that the additional overhead of sampling the subset in GPU is negligible.

Table 4: Comparison of time (second) spends in each stage of training 3-layer GCN on `Reddit` dataset using different sampling techniques.

|  |  | **Sampling** | **Transfer** | **Computation** | **Overhead** |
|---|---|---|---|---|---|
| **Reddit** | Nodewise | 845.21 | 22.73 | 13.13 | - |
|  | Nodewise+LG | 25.76 | 0.76 | 11.69 | 0.04 |
|  | Layerwise | 9.12 | 2.14 | 1.04 | - |
|  | Layerwise+LG | 0.56 | 0.06 | 0.98 | 0.04 |
|  | Subgraph | 4.66 | 1.58 | 0.98 | - |
|  | Subgraph+LG | 0.19 | 0.06 | 0.96 | 0.04 |

# B  Summary of notations

We summarize all notations used throughout the paper in Table 5. Furthermore, we introduce the necessary notations for our theoretical analysis. First, we introduce the **full-batch GCN** which is defined as gradient calculation using all inner nodes and full-batch as proposed in [15]. In addition, we discuss **sampling-based GCN** which uses mini-batches for gradient calculation, and samples a subset of neighbors to estimate the feature representation for nodes in the mini-batch. An example of sampling-based GCN is [13]. Moreover, we introduce **mini-batch GCN**, which is defined as using mini-batch for gradient calculation, but using all the neighbors to calculate the exact feature representation for nodes in the mini-batch. Mini-batch GCN can be regarded as an intermediate schema between the full-batch GCN and the sampling-based GCN, which is used for the the mean-square error decomposition of stochastic gradient.

Table 5: Summary of notations used in this paper

| | |
|---|---|
| $\mathcal{G} = (\mathcal{V}, \mathcal{E})$ | $\mathcal{G}$ denotes the graph consist of set of $N = |\mathcal{V}|$ nodes and $M = |\mathcal{E}|$ edges. |
| $\mathbf{A}$ | Denotes the adjacency matrix corresponding to the graph $\mathcal{G}$. |
| $\mathbf{D}$ | Denotes the degree matrix corresponding to the graph $\mathcal{G}$. |
| $\mathbf{L}, \widetilde{\mathbf{L}}^{(\ell)}$ | $\mathbf{L}$ denotes the full Laplacian matrix calculated by $\mathbf{L} = \mathbf{D}^{-1/2}\mathbf{A}\mathbf{D}^{-1/2}$ and $\widetilde{\mathbf{L}}^{(\ell)}$ is a sampled Laplacian matrix sampled from $\mathbf{L}$ for the $\ell$th layer. |
| $\mathbf{X}, \boldsymbol{x}_i$ | $\mathbf{X}$ denotes the node feature matrix for all $N$ nodes where $\mathbf{X} = [\boldsymbol{x}_1, \ldots, \boldsymbol{x}_N]$. |
| $\boldsymbol{y}_i$ | Denotes the label vector in $\mathbb{R}^C$. For example, in binary classification task, $\boldsymbol{y}_i \in \mathbb{R}^C$ is a one-hot vector with $C = 2$. |
| $\mathbf{W}^{(\ell)}, \boldsymbol{\theta}, \boldsymbol{\theta}_{k,r}$ | $\mathbf{W}^{(\ell)}$ denotes the weight matrix for the $\ell$th graph convolution layer, $\boldsymbol{\theta} := \{\mathbf{W}^{(1)}, \ldots, \mathbf{W}^{(L)}\}$ denotes the stacked parameters, and $\boldsymbol{\theta}_{k,r}$ denotes the weight for the $r$th iteration in the $k$th recycling epoch. |
| $\mathbf{H}^{(\ell)}, \mathbf{Z}^{(\ell)}$ | $\mathbf{Z}^{(\ell)}$ denotes the node feature matrix for the $\ell$th layer before activation and $\mathbf{H}^{(\ell)} = \sigma(\mathbf{Z}^{(\ell)})$ denotes the node feature matrix after the $\ell$th layer. |
| $\mathcal{V}_k^{(\ell)}$ | Denotes the $\ell$ layer nodes sampled at the beginning of each recycling epoch. |
| $\mathcal{V}_{k,r}$ | Denotes a subset of nodes sampled from $\mathcal{V}_k^{(L)}$ inside the $k$th recycling epoch. |
| $S, B$ | $S = |\mathcal{V}_k^{(L)}|$ and $B = |\mathcal{V}_{k,r}|$. |
| $D, \widetilde{D}$ | Suppose all nodes have the same number of neighbors. $D$ is the number of neighbors for each node and $\widetilde{D}$ is the number of neighbors sampled during training. |
| $K, R, T, \rho$ | $K$ denotes the number of recycling epoch, $T = \sum_{k=1}^{K} \rho^k R$ is the number of iterations for training, and $\rho$ and $R$ are parameters that control the size of recycling epoch. |
| $\sigma(\cdot)$ | $\sigma(\cdot)$ is the activation function, e.g., ReLU function. |
| $\phi(\cdot, \boldsymbol{y}_i)$ | $\phi(\cdot, \boldsymbol{y}_i)$ is the loss function, e.g., cross-entropy loss $$\phi(\boldsymbol{z}_i, \boldsymbol{y}_i) = -\log\left(\frac{\exp(\boldsymbol{z}_i^\top \boldsymbol{y}_i)}{\sum_{j=1}^{C} \exp([\boldsymbol{z}_i]_j))}\right),$$ where label vector $\boldsymbol{y}_i$ is a one-hot vector. |

**Full-batch GCN.** Suppose we are solving multi-class node classification problem. Given $\phi(\boldsymbol{z}_i, \boldsymbol{y}_i)$ as the loss function (e.g., cross-entropy loss), $\boldsymbol{z}_i \in \mathbb{R}^C$ as the final prediction of node $i$, and $\boldsymbol{y}_i \in \mathbb{R}^C$ as the label of node $i$. Let $\boldsymbol{\theta} = \{\mathbf{W}\}$, then the loss function of full-batch GCN is defined as

$$\mathcal{L}(\boldsymbol{\theta}) = \frac{1}{N} \sum_{i \in \mathcal{V}} \phi(\boldsymbol{z}_i, \boldsymbol{y}_i), \ \boldsymbol{z}_i = \sum_{j \in \mathcal{N}(i)} \mathbf{L}_{i,j}(\mathbf{W}^\top \boldsymbol{x}_j) \tag{9}$$

where Laplacian matrix is defined as $\mathbf{L} = \mathbf{D}^{-1}\mathbf{A}$. Its corresponding objective function is defined as

$$F(\boldsymbol{\theta}) = \frac{1}{N} \sum_{i \in \mathcal{V}} f_i\left(\frac{1}{|\mathcal{N}(i)|} \sum_{j \in \mathcal{N}(i)} g_j(\boldsymbol{\theta})\right), \tag{10}$$

and its gradient is calculated by

$$\nabla F(\boldsymbol{\theta}) = \frac{1}{N} \sum_{i \in \mathcal{V}} \nabla f_i\left(\frac{1}{|\mathcal{N}(i)|} \sum_{j \in \mathcal{N}(i)} g_j(\boldsymbol{\theta})\right)\left(\frac{1}{|\mathcal{N}(i)|} \sum_{j \in \mathcal{N}(i)} \nabla g_j(\boldsymbol{\theta})\right). \tag{11}$$

**Sampling-based GCN.** The loss function of sampling-based GCN is defined as

$$\widetilde{\mathcal{L}}(\boldsymbol{\theta}) = \frac{1}{B} \sum_{i \in \mathcal{V}_\mathcal{B}} \phi(\widetilde{\boldsymbol{z}}_i, \boldsymbol{y}_i), \ \widetilde{\boldsymbol{z}}_i = \sum_{j \in \widetilde{\mathcal{N}}(i)} \widetilde{\mathbf{L}}_{i,j}(\mathbf{W}^\top \boldsymbol{x}_j), \tag{12}$$

where $\widetilde{\mathbf{L}}_{i,j} = (|\mathcal{N}(i)|/|\widetilde{\mathcal{N}}(i)|)\mathbf{L}_{i,j}$ for $j \in \widetilde{\mathcal{N}}(i)$ otherwise $\widetilde{\mathbf{L}}_{i,j} = 0$. The stochastic gradient that used to update the parameter is computed as

$$\nabla \widetilde{F}_{\mathcal{B}}(\boldsymbol{\theta}) = \frac{1}{B} \sum_{i \in \mathcal{V}_{\mathcal{B}}} \nabla f_i \Big( \frac{1}{|\widetilde{\mathcal{N}}(i)|} \sum_{j \in \widetilde{\mathcal{N}}(i)} g_j(\boldsymbol{\theta}) \Big) \Big( \frac{1}{|\widetilde{\mathcal{N}}(i)|} \sum_{j \in \widetilde{\mathcal{N}}(i)} \nabla g_j(\boldsymbol{\theta}) \Big) \tag{13}$$

**Mini-batch GCN.** The loss function of mini-batch GCN is defined as

$$\mathcal{L}(\boldsymbol{\theta}) = \frac{1}{B} \sum_{i \in \mathcal{V}_{\mathcal{B}}} \phi(\boldsymbol{z}_i, \boldsymbol{y}_i), \ \boldsymbol{z}_i = \sum_{j \in \mathcal{N}(i)} \mathbf{L}_{i,j}(\mathbf{W}^\top \boldsymbol{x}_j), \tag{14}$$

and the stochastic gradient that used to update the parameter is computed as

$$\nabla F_{\mathcal{B}}(\boldsymbol{\theta}) = \frac{1}{B} \sum_{i \in \mathcal{V}_{\mathcal{B}}} \nabla f_i \Big( \frac{1}{|\mathcal{N}(i)|} \sum_{j \in \mathcal{N}(i)} g_j(\boldsymbol{\theta}) \Big) \Big( \frac{1}{|\mathcal{N}(i)|} \sum_{j \in \mathcal{N}(i)} \nabla g_j(\boldsymbol{\theta}) \Big) \tag{15}$$

## C  Proof of Theorem 1

The proof of Theorem 1 follows from the standard stochastic non-convex optimization, with the key difference that unlike vanilla SGD where the stochastic gradient is an unbiased estimation of full gradient, in sampling-based GCNs, the stochastic gradient is biased and the bias is controlled by the number of neighbors sampled during training.

To ease the proof of the Theorem 1, we state few lemmas. We begin by the following lemma that gives the Lipschitz continuous constant of the gradient of composite function $F(\boldsymbol{\theta})$, which plays an important role in choosing the learning rate and the final convergence rate.

**Lemma 3.** *Under Assumption 1, the gradient of $F(\boldsymbol{\theta})$ is Lipschitz continuous with constant*

$$L_F = L_f G_g + L_g^2 G_f. \tag{16}$$

The following lemma shows the mean-square error of stochastic gradients, which is the key factor in convergence and leads to slow convergence rate when the mean-square error is large.

**Lemma 4.** *Suppose objective function $F(\boldsymbol{\theta})$ has $L_F$-Lipschitz continuous gradient and the expected mean-square error of stochastic gradient $\widetilde{F}_{\mathcal{B}}(\boldsymbol{\theta}_t)$ to the full gradient $\nabla F(\boldsymbol{\theta}_t)$ is defined as*

$$\Delta = \frac{1}{T} \sum_{t=1}^{T} \mathbb{E} \left[ \left\| \nabla \widetilde{F}_{\mathcal{B}}(\boldsymbol{\theta}_t) - \nabla F(\boldsymbol{\theta}_t) \right\|^2 \right]. \tag{17}$$

*Set step size $\eta = \min \left\{ \frac{3}{2L_F}, \sqrt{\frac{\mathbb{E}[F(\boldsymbol{\theta}_1)] - \mathbb{E}[F(\boldsymbol{\theta}_\star)]}{L_F \Delta T}} \right\}$ and for $\tilde{\boldsymbol{\theta}} = \min_t \mathbb{E} \left[ \|\nabla F(\boldsymbol{\theta}_t)\| \right]$, then we have*

$$\mathbb{E}[\|\nabla F(\tilde{\boldsymbol{\theta}})\|^2] \leq \mathcal{O}\left( \sqrt{\Delta/T} \right). \tag{18}$$

Next, we show that the mean-square error of stochastic gradient in sampling-based GCN can be decomposed into feature approximation error $\text{Err}_1(t)$ and gradient approximation error $\text{Err}_2(t)$.

$$\begin{aligned} \Delta := \frac{1}{T} \sum_{t=1}^{T} \mathbb{E} \left[ \left\| \nabla \widetilde{F}_{\mathcal{B}}(\boldsymbol{\theta}_t) - \nabla F(\boldsymbol{\theta}_t) \right\|^2 \right] \\ \leq \frac{2}{T} \sum_{t=1}^{T} \underbrace{\mathbb{E} \left[ \left\| \nabla \widetilde{F}_{\mathcal{B}}(\boldsymbol{\theta}_t) - \nabla F_{\mathcal{B}}(\boldsymbol{\theta}_t) \right\|^2 \right]}_{\text{Err}_1(t)} + \frac{2}{T} \sum_{t=1}^{T} \underbrace{\mathbb{E} \left[ \|\nabla F_{\mathcal{B}}(\boldsymbol{\theta}_t) - \nabla F(\boldsymbol{\theta}_t)\|^2 \right]}_{\text{Err}_2(t)}, \end{aligned} \tag{19}$$

where the inequality hold due to $\|\boldsymbol{x} + \boldsymbol{y}\|^2 \leq 2\|\boldsymbol{x}\|^2 + 2\|\boldsymbol{y}\|^2$ for any $\boldsymbol{x}, \boldsymbol{y}$. Then, in Lemma 5, we provide an upper bound on the feature approximation error $\text{Err}_1(t)$ and, in Lemma 6, we provide an upper bound on the gradient approximation error $\text{Err}_2(t)$. The proof are based on matrix concentration.

**Lemma 5.** *Denote $\widetilde{D}$ as the number of selected neighbors in the sampling and $d$ as the dimension of node features. With probability at least $1 - \delta$ we have*

$$\frac{1}{2}\mathbb{E}\left[\left\|\nabla\widetilde{F}_{\mathcal{B}}(\boldsymbol{\theta}) - \nabla F_{\mathcal{B}}(\boldsymbol{\theta})\right\|^2\right] \leq 16L_f^2 G_g^2 \frac{\log(2d/\delta)}{\widetilde{D}} + 32G_f^2 L_g^4 \frac{\log(2d/\delta) + 1/4}{\widetilde{D}}. \tag{20}$$

**Lemma 6.** *Denote $B$ as the size of mini-batch and $d$ as the dimension of node features. With probability at least $1 - \delta$ we have*

$$\frac{1}{2}\mathbb{E}\left[\|\nabla F_{\mathcal{B}}(\boldsymbol{\theta}) - \nabla F(\boldsymbol{\theta})\|^2\right] \leq 32G_f^2 \frac{\log(2d/\delta) + 1/4}{B}. \tag{21}$$

By plugging back the results in Lemma 5 and Lemma 6 back to Eq. 19, we have

$$\Delta = \mathcal{O}\left(L_f^2 G_g^2 \frac{\log(2d/\delta)}{\widetilde{D}}\right) + \mathcal{O}\left(G_f^2 L_g^4 \frac{\log(2d/\delta) + 1/4}{\widetilde{D}}\right) + \mathcal{O}\left(G_f^2 \frac{\log(2d/\delta) + 1/4}{B}\right). \tag{22}$$

From Lemma 3 we know that

$$L_F = L_f G_g + L_g^2 G_f. \tag{23}$$

Plugging the result back to Lemma 4 concludes the proof.

## C.1 Proof of Lemma 3

We say function $F(\boldsymbol{\theta})$ has $L_F$-Lipschitz continuous gradient if for any $\boldsymbol{\theta}_1$ and $\boldsymbol{\theta}_2$, we have the following inequality

$$\|\nabla F(\boldsymbol{\theta}_1) - \nabla F(\boldsymbol{\theta}_2)\| \leq L_F \|\boldsymbol{\theta}_1 - \boldsymbol{\theta}_2\|. \tag{24}$$

Our next step is to compute $L_F$. Denote $\nabla F(\boldsymbol{\theta}_1)$ as

$$\nabla F(\boldsymbol{\theta}_1) = \frac{1}{N}\sum_{i\in\mathcal{V}}\underbrace{\nabla f_i\left(\frac{1}{D}\sum_{j\in\mathcal{N}(i)}g_j(\boldsymbol{\theta}_1)\right)}_{A_1(i)}\underbrace{\left(\frac{1}{D}\sum_{j\in\mathcal{N}(i)}\nabla g_j(\boldsymbol{\theta}_1)\right)}_{A_2(i)}. \tag{25}$$

and $\nabla F(\boldsymbol{\theta}_2)$ as

$$\nabla F(\boldsymbol{\theta}_2) = \frac{1}{N}\sum_{i\in\mathcal{V}}\underbrace{\nabla f_i\left(\frac{1}{D}\sum_{j\in\mathcal{N}(i)}g_j(\boldsymbol{\theta}_2)\right)}_{B_1(i)}\underbrace{\left(\frac{1}{D}\sum_{j\in\mathcal{N}(i)}\nabla g_j(\boldsymbol{\theta}_2)\right)}_{B_2(i)}. \tag{26}$$

Then we have

$$
\begin{aligned}
\|\nabla F(\boldsymbol{\theta}_1) - \nabla F(\boldsymbol{\theta}_2)\| &= \left\|\frac{1}{N}\sum_{i\in\mathcal{V}}A_1(i)A_2(i) - \frac{1}{N}\sum_{i\in\mathcal{V}}B_1(i)B_2(i)\right\| \\
&= \left\|\frac{1}{N}\sum_{i\in\mathcal{V}}A_1(i)\left(A_2(i) - B_2(i)\right) + \frac{1}{N}\sum_{i\in\mathcal{V}}\left(A_1(i) - B_1(i)\right)B_2(i)\right\| \\
&\underset{(a)}{\leq} \frac{1}{N}\sum_{i\in\mathcal{V}}\|A_1(i)\|\|A_2(i) - B_2(i)\| + \frac{1}{N}\sum_{i\in\mathcal{V}}\|A_1(i) - B_1(i)\|\|B_2(i)\| \\
&\underset{(b)}{\leq} \left(L_f G_g + L_g^2 G_f\right)\|\boldsymbol{\theta}_1 - \boldsymbol{\theta}_2\|,
\end{aligned}
$$
$$\tag{27}$$

where the inequality (a) is due to $\|\boldsymbol{x} + \boldsymbol{y}\| \leq \|\boldsymbol{x}\| + \|\boldsymbol{y}\|$ and $\|\boldsymbol{xy}\| \leq \|\boldsymbol{x}\|\|\boldsymbol{y}\|$ for any $\boldsymbol{x}, \boldsymbol{y}$, and the inequality (b) is due to the Assumption 1.

## C.2 Proof of Lemma 4

For the ease of presentation, let us denote stochastic gradient as $\boldsymbol{g}_t := \nabla \widetilde{F}_{\mathcal{B}}(\boldsymbol{\theta}_t)$ and full gradient as $\nabla F(\boldsymbol{\theta}_t)$.

By the update rule $\boldsymbol{\theta}_{t+1} = \boldsymbol{\theta}_t - \eta \boldsymbol{g}_t$ we obtain,

$$
\begin{aligned}
F(\boldsymbol{\theta}_{t+1}) - F(\boldsymbol{\theta}_t) &\leq \langle \nabla F(\boldsymbol{\theta}_t), \boldsymbol{\theta}_{t+1} - \boldsymbol{\theta}_t \rangle + \frac{L_F}{2} \|\boldsymbol{\theta}_{t+1} - \boldsymbol{\theta}_t\|^2 \\
&= -\eta \langle \nabla F(\boldsymbol{\theta}_t), \boldsymbol{g}_t \rangle + \frac{L_F}{2} \|\boldsymbol{\theta}_{t+1} - \boldsymbol{\theta}_t\|^2 \\
&= -\eta \langle \nabla F(\boldsymbol{\theta}_t), \nabla F(\boldsymbol{\theta}_t) - \nabla F(\boldsymbol{\theta}_t) + \boldsymbol{g}_t \rangle + \frac{L_F}{2} \|\boldsymbol{\theta}_{t+1} - \boldsymbol{\theta}_t\|^2 \\
&= -\eta \|\nabla F(\boldsymbol{\theta}_t)\|^2 - \eta \langle \nabla F(\boldsymbol{\theta}_t), \boldsymbol{g}_t - \nabla F(\boldsymbol{\theta}_t) \rangle + \frac{\eta^2 L_F}{2} \|\boldsymbol{g}_t\|^2.
\end{aligned}
\tag{28}
$$

Adding and subtracting $\nabla F(\boldsymbol{\theta}_t)$ to $\boldsymbol{g}_t$ gives

$$
\begin{aligned}
\|\boldsymbol{g}_t\|^2 &= \|\boldsymbol{g}_t - \nabla F(\boldsymbol{\theta}_t) + \nabla F(\boldsymbol{\theta}_t)\|^2 \\
&= \|\boldsymbol{g}_t - \nabla F(\boldsymbol{\theta}_t)\|^2 + \|\nabla F(\boldsymbol{\theta}_t)\|^2 + 2\langle \nabla F(\boldsymbol{\theta}_t), \boldsymbol{g}_t - \nabla F(\boldsymbol{\theta}_t) \rangle.
\end{aligned}
\tag{29}
$$

By plugging the above equality back, we get

$$
\begin{aligned}
F(\boldsymbol{\theta}_{t+1}) - F(\boldsymbol{\theta}_t) &\leq \left( \frac{\eta^2 L_F}{2} - \eta \right) \|\nabla F(\boldsymbol{\theta}_t)\|^2 + \left( \eta^2 L_F - \eta \right) \langle \nabla F(\boldsymbol{\theta}_t), \boldsymbol{g}_t - \nabla F(\boldsymbol{\theta}_t) \rangle \\
&\quad + \frac{\eta^2 L_F}{2} \|\boldsymbol{g}_t - \nabla F(\boldsymbol{\theta}_t)\|^2.
\end{aligned}
\tag{30}
$$

By using the fact that $2\langle \nabla F(\boldsymbol{\theta}_t), \boldsymbol{g}_t - \nabla F(\boldsymbol{\theta}_t) \rangle \leq \|\nabla F(\boldsymbol{\theta}_t)\|^2 + \|\boldsymbol{g}_t - \nabla F(\boldsymbol{\theta}_t)\|^2$, we have

$$
F(\boldsymbol{\theta}_{t+1}) - F(\boldsymbol{\theta}_t) \leq \left( \eta^2 L_F - \frac{3}{2}\eta \right) \|\nabla F(\boldsymbol{\theta}_t)\|^2 + \left( \eta^2 L_F - \frac{1}{2}\eta \right) \|\boldsymbol{g}_t - \nabla F(\boldsymbol{\theta}_t)\|^2.
\tag{31}
$$

Taking expectation on both side and rearranging the terms results in

$$
\begin{aligned}
\left( \frac{3}{2}\eta - \eta^2 L_F \right) \mathbb{E}[\|\nabla F(\boldsymbol{\theta}_t)\|^2] &\leq \mathbb{E}[F(\boldsymbol{\theta}_t)] - \mathbb{E}[F(\boldsymbol{\theta}_{t+1})] + \left( \eta^2 L_F - \frac{1}{2}\eta \right) \mathbb{E}[\|\boldsymbol{g}_t - \nabla F(\boldsymbol{\theta}_t)\|^2] \\
&\leq \mathbb{E}[F(\boldsymbol{\theta}_t)] - \mathbb{E}[F(\boldsymbol{\theta}_{t+1})] + \eta^2 L_F \mathbb{E}[\|\boldsymbol{g}_t - \nabla F(\boldsymbol{\theta}_t)\|^2].
\end{aligned}
\tag{32}
$$

By summing up above inequality for all $T$ iterations, denoting $\Delta = \frac{1}{T} \sum_{t=1}^{T} \mathbb{E}[\|\boldsymbol{g}_t - \nabla F(\boldsymbol{\theta}_t)\|^2]$, and using the fact that $F(\boldsymbol{\theta}_\star) \leq F(\boldsymbol{\theta}_{t+1})$ gives

$$
\left( \frac{3}{2}\eta - \eta^2 L_F \right) \sum_{t=1}^{T} \mathbb{E}[\|\nabla F(\boldsymbol{\theta}_t)\|^2] \leq \mathbb{E}[F(\boldsymbol{\theta}_1)] - \mathbb{E}[F(\boldsymbol{\theta}_\star)] + T\eta^2 L_F \Delta.
\tag{33}
$$

Dividing both side by $T \left( \frac{3}{2}\eta - \eta^2 L_F \right)$ results in

$$
\begin{aligned}
\frac{1}{T} \sum_{t=1}^{T} \mathbb{E}[\|\nabla F(\boldsymbol{\theta}_t)\|^2] &\leq \left( \frac{3}{2}\eta - \eta^2 L_F \right)^{-1} \frac{(\mathbb{E}[F(\boldsymbol{\theta}_1)] - \mathbb{E}[F(\boldsymbol{\theta}_\star)])}{T} + \Delta \frac{2\eta^2 L_F}{(3\eta - 2\eta^2 L_F)} \\
&= \left( 3\eta - 2\eta^2 L_F \right)^{-1} \frac{2\left(\mathbb{E}[F(\boldsymbol{\theta}_1)] - \mathbb{E}[F(\boldsymbol{\theta}_\star)]\right)}{T} + \Delta \frac{2\eta L_F}{(3 - 2\eta L_F)}.
\end{aligned}
\tag{34}
$$

By choosing $\eta = \min\left\{ \frac{3}{2L_F}, \sqrt{\frac{\mathbb{E}[F(\boldsymbol{\theta}_1)] - \mathbb{E}[F(\boldsymbol{\theta}_\star)]}{\Delta L_F T}} \right\}$ we have

$$
\begin{aligned}
\frac{1}{T} \sum_{t=1}^{T} \mathbb{E}[\|\nabla F(\boldsymbol{\theta}_t)\|^2] &\leq \frac{1}{\eta\,(3 - 2\eta L_F)} \frac{2\left(\mathbb{E}[F(\boldsymbol{\theta}_1)] - \mathbb{E}[F(\boldsymbol{\theta}_\star)]\right)}{T} + \Delta \frac{2\eta L_F}{(3 - 2\eta L_F)} \\
&\leq \frac{1}{\eta} \frac{2\left(\mathbb{E}[F(\boldsymbol{\theta}_1)] - \mathbb{E}[F(\boldsymbol{\theta}_\star)]\right)}{T} + 2\Delta \eta L_F \\
&\leq \mathcal{O}\left( \sqrt{\Delta/T} \right).
\end{aligned}
\tag{35}
$$

Since $\tilde{\boldsymbol{\theta}}$ is decided in a way that has minimum gradient, it follows

$$\mathbb{E}[\|\nabla F(\tilde{\boldsymbol{\theta}})\|^2] \leq \mathcal{O}\left(\sqrt{\Delta/T}\right), \tag{36}$$

which gives the bound as stated in the lemma.

### C.3 Proof of Lemma 5

By definition, we can bound $\mathbb{E}[\|\nabla \widetilde{F}_{\mathcal{B}}(\boldsymbol{\theta}) - \nabla F_{\mathcal{B}}(\boldsymbol{\theta})\|^2]$ by adding and subtracting intermediate terms inside such that each adjacent pair of products differ at most in one factor. To do so, recall the definitions of $\nabla F_{\mathcal{B}}(\boldsymbol{\theta})$ and $\nabla \widetilde{F}_{\mathcal{B}}(\boldsymbol{\theta})$ as

$$\nabla F_{\mathcal{B}}(\boldsymbol{\theta}) = \frac{1}{B} \sum_{i \in \mathcal{V}_{\mathcal{B}}} \underbrace{\nabla f_i \left(\frac{1}{D} \sum_{j \in \mathcal{N}(i)} g_j(\boldsymbol{\theta})\right)}_{A_1(i)} \underbrace{\left(\frac{1}{D} \sum_{j \in \mathcal{N}(i)} \nabla g_j(\boldsymbol{\theta})\right)}_{A_2(i)}, \tag{37}$$

and

$$\nabla \widetilde{F}_{\mathcal{B}}(\boldsymbol{\theta}) = \frac{1}{B} \sum_{i \in \mathcal{V}_{\mathcal{B}}} \underbrace{\nabla f_i \left(\frac{1}{\widetilde{D}} \sum_{j \in \widetilde{\mathcal{N}}(i)} g_j(\boldsymbol{\theta})\right)}_{B_1(i)} \underbrace{\left(\frac{1}{\widetilde{D}} \sum_{j \in \widetilde{\mathcal{N}}(i)} \nabla g_j(\boldsymbol{\theta})\right)}_{B_2(i)}. \tag{38}$$

For simplicity, we denote $\mathbb{E}_{\mathcal{V}_{\mathcal{B}} \sim \mathcal{V}} \left[\mathbb{E}_{j \sim \mathcal{N}(i), \forall i \in \mathcal{V}_{\mathcal{B}}}[\cdot]\right]$ as $\mathbb{E}[\cdot]$. We have

$$\begin{aligned}
&\mathbb{E}\left[\left\|\nabla \widetilde{F}_{\mathcal{B}}(\boldsymbol{\theta}) - \nabla F_{\mathcal{B}}(\boldsymbol{\theta})\right\|^2\right] \\
&= \mathbb{E}\left[\left\|\frac{1}{B}\sum_{i \in \mathcal{V}_{\mathcal{B}}} B_1(i)B_2(i) - \frac{1}{B}\sum_{i \in \mathcal{V}_{\mathcal{B}}} A_1(i)A_2(i)\right\|^2\right] \\
&\underset{(a)}{\leq} \mathbb{E}\left[\|B_1(i)B_2(i) - A_1(i)A_2(i)\|^2\right] \\
&\underset{(b)}{\leq} 2\mathbb{E}\left[\|B_1(i)(B_2(i) - A_2(i))\|^2\right] + 2\mathbb{E}\left[\|(B_1(i) - A_1(i))A_2(i)\|^2\right] \\
&\underset{(c)}{\leq} 2\mathbb{E}\left[\|B_1(i)\|^2\right] \mathbb{E}\left[\|B_2(i) - A_2(i)\|^2\right] + 2\mathbb{E}\left[\|B_1(i) - A_1(i)\|^2\right] \mathbb{E}\left[\|A_2(i)\|^2\right],
\end{aligned} \tag{39}$$

where inequality (a) is due to $\|\frac{1}{n}\sum_{i=1}^n \boldsymbol{x}_i\| \leq \frac{1}{n}\sum_{i=1}^n \|\boldsymbol{x}_i\|$, inequality (b) is due to $\|\boldsymbol{x} + \boldsymbol{y}\|^2 \leq 2\|\boldsymbol{x}\| + 2\|\boldsymbol{y}\|$, and inequality (c) is due to $\|\boldsymbol{x}\boldsymbol{y}\| \leq \|\boldsymbol{x}\|\|\boldsymbol{y}\|$.

(1) Considering $\mathbb{E}\left[\|B_1(i)\|^2\right]$, we have

$$\mathbb{E}\left[\|B_1(i)\|^2\right] = \mathbb{E}\left[\left\|\nabla f_i \left(\frac{1}{\widetilde{D}} \sum_{j \in \widetilde{\mathcal{N}}(i)} g_j(\boldsymbol{\theta})\right)\right\|^2\right] \leq L_f^2. \tag{40}$$

(2) Considering $\mathbb{E}\left[\|A_2(i)\|^2\right]$, we have

$$\mathbb{E}\left[\|A_2(i)\|^2\right] = \mathbb{E}\left[\left\|\frac{1}{D} \sum_{j \in \mathcal{N}(i)} \nabla g_j(\boldsymbol{\theta})\right\|^2\right] \leq L_g^2. \tag{41}$$

(3) Considering $\mathbb{E}\left[\|B_1(i) - A_1(i)\|^2\right]$, we have

$$
\begin{aligned}
\mathbb{E}\left[\|B_1(i) - A_1(i)\|^2\right] &= \mathbb{E}\left[\left\|\nabla f_i\left(\frac{1}{\widetilde{D}}\sum_{j\in\widetilde{\mathcal{N}}(i)} g_j(\boldsymbol{\theta})\right) - \nabla f_i\left(\frac{1}{D}\sum_{j\in\mathcal{N}(i)} g_j(\boldsymbol{\theta})\right)\right\|^2\right] \\
&\leq G_f^2\mathbb{E}\left[\left\|\frac{1}{\widetilde{D}}\sum_{j\in\widetilde{\mathcal{N}}(i)} g_j(\boldsymbol{\theta}) - \frac{1}{D}\sum_{j\in\mathcal{N}(i)} g_j(\boldsymbol{\theta})\right\|^2\right] \\
&\leq 32G_f^2 L_g^2\frac{\log(2d/\delta) + 1/4}{\widetilde{D}},
\end{aligned}
\tag{42}
$$

where the last inequality is due to Lemma 12.

(4) Considering $\mathbb{E}\left[\|B_2(i) - A_2(i)\|^2\right]$, we have

$$
\begin{aligned}
\mathbb{E}\left[\|B_2(i) - A_2(i)\|^2\right] &= \mathbb{E}\left[\left\|\frac{1}{\widetilde{D}}\sum_{j\in\widetilde{\mathcal{N}}(i)} \nabla g_j(\boldsymbol{\theta}) - \frac{1}{D}\sum_{j\in\mathcal{N}(i)} \nabla g_j(\boldsymbol{\theta})\right\|^2\right] \\
&\leq 16G_g^2\frac{\log(2d/\delta)}{\widetilde{D}},
\end{aligned}
\tag{43}
$$

where the last inequality is due to Lemma 12.

(6) By plugging Eq. 40, Eq. 41, Eq. 42, and Eq. 43 back to Eq. 39, with probability at least $1-\delta$, it holds that:

$$
\mathbb{E}\left[\left\|\nabla\widetilde{F}_{\mathcal{B}}(\boldsymbol{\theta}) - \nabla F_{\mathcal{B}}(\boldsymbol{\theta})\right\|^2\right] \leq 32L_f^2 G_g^2\frac{\log(2d/\delta)}{\widetilde{D}} + 64G_f^2 L_g^4\frac{\log(2d/\delta) + 1/4}{\widetilde{D}}.
\tag{44}
$$

### C.4 Proof of Lemma 6

By definition, we know $\nabla F_{\mathcal{B}}(\boldsymbol{\theta})$ and $\nabla F(\boldsymbol{\theta})$ are defined as

$$
\nabla F_{\mathcal{B}}(\boldsymbol{\theta}) = \frac{1}{B}\sum_{i\in\mathcal{V}_{\mathcal{B}}} \underbrace{\nabla f_i\left(\frac{1}{D}\sum_{j\in\mathcal{N}(i)} g_j(\boldsymbol{\theta})\right)}_{A_1(i)}\underbrace{\left(\frac{1}{D}\sum_{j\in\mathcal{N}(i)} \nabla g_j(\boldsymbol{\theta})\right)}_{A_2(i)},
\tag{45}
$$

$$
\nabla F(\boldsymbol{\theta}) = \frac{1}{N}\sum_{i\in\mathcal{V}} \underbrace{\nabla f_i\left(\frac{1}{D}\sum_{j\in\mathcal{N}(i)} g_j(\boldsymbol{\theta})\right)}_{A_1(i)}\underbrace{\left(\frac{1}{D}\sum_{j\in\mathcal{N}(i)} \nabla g_j(\boldsymbol{\theta})\right)}_{A_2(i)}.
\tag{46}
$$

For simplicity, we denote $\mathbb{E}_{\mathcal{V}_{\mathcal{S}}\sim\mathcal{V}}\left[\mathbb{E}_{j\sim\mathcal{N}(i),\forall i\in\mathcal{V}_{\mathcal{B}} \mid \mathcal{V}_{\mathcal{B}}\sim\mathcal{V}_{\mathcal{S}}}[\cdot]\right]$ as $\mathbb{E}[\cdot]$.

$$
\begin{aligned}
\mathbb{E}\left[\|\nabla F_{\mathcal{B}}(\boldsymbol{\theta}) - \nabla F(\boldsymbol{\theta})\|^2\right] &\leq \mathbb{E}\left[\left\|\frac{1}{B}\sum_{i\in\mathcal{V}_{\mathcal{B}}} A_1(i)A_2(i) - \frac{1}{N}\sum_{i\in\mathcal{V}} A_1(i)A_2(i)\right\|^2\right] \\
&\leq 32G_f^2\frac{\log(2d/\delta) + 1/4}{B},
\end{aligned}
\tag{47}
$$

where the last inequality is due to Lemma 12.

# D  Proof of Theorem 2

The proof of the Theorem 2 is similar to the proof of Theorem 1, with a key difference: unlike vanilla sampling-based GCN that samples mini-batch nodes and inner-layer nodes every iteration, in LAZYGCN, we only perform mini-batch nodes and inner-layer nodes sampling at the beginning of each recycling epoch, then sample mini-batch nodes with fixed inner layer nodes during the recycling stage. The proof of Theorem 2 is based on the following lemmas.

The following lemma shows that the mean-square error of stochastic gradient is the key factor in convergence and leads to slow convergence rate when the mean-sqaure error is large.

**Lemma 7.** *Suppose objective function $F(\boldsymbol{\theta})$ has $L_F$-Lipschitz continuous gradient and the expected mean-square error of stochastic gradient $\widetilde{F}_{\mathcal{B}}(\boldsymbol{\theta}_t)$ to the full gradient $\nabla F(\boldsymbol{\theta}_t)$ is bounded by*

$$\Delta = \frac{1}{T} \sum_{t=1}^{T} \mathbb{E}\left[\left\|\widetilde{F}_{\mathcal{B}}(\boldsymbol{\theta}_t) - \nabla F(\boldsymbol{\theta}_t)\right\|^2\right]. \tag{48}$$

*By choosing step size $\eta = \min\left\{\frac{3}{2L_F}, \sqrt{\frac{\mathbb{E}[F(\boldsymbol{\theta}_1)] - \mathbb{E}[F(\boldsymbol{\theta}_\star)]}{L_F(\sum_{k=1}^{K} \rho^k R \Delta_k)}}\right\}$, for $\tilde{\boldsymbol{\theta}} = \min_t \mathbb{E}\left[\|\nabla F(\boldsymbol{\theta}_t)\|\right]$ we have*

$$\mathbb{E}[\|\nabla F(\tilde{\boldsymbol{\theta}})\|^2] \leq \mathcal{O}\left(\frac{\sqrt{\sum_{k=1}^{K} \rho^k R \Delta_k}}{T}\right). \tag{49}$$

Similar to the analysis of vanilla sampling-based GCN training, the mean-square error of stochastic gradient in LAZYGCN can also be decomposed into feature approximation error $\mathrm{Err}_1(t)$ and gradient approximation error $\mathrm{Err}_2(t)$.

$$\Delta := \frac{1}{T} \sum_{t=1}^{T} \mathbb{E}\left[\left\|\nabla \widetilde{F}_{\mathcal{B}}(\boldsymbol{\theta}_t) - \nabla F(\boldsymbol{\theta}_t)\right\|^2\right]$$

$$\leq \frac{2}{T} \sum_{t=1}^{T} \underbrace{\mathbb{E}\left[\left\|\nabla \widetilde{F}_{\mathcal{B}}(\boldsymbol{\theta}_t) - \nabla F_{\mathcal{B}}(\boldsymbol{\theta}_t)\right\|^2\right]}_{\mathrm{Err}_1(t)} + \frac{2}{T} \sum_{t=1}^{T} \underbrace{\mathbb{E}\left[\|\nabla F_{\mathcal{B}}(\boldsymbol{\theta}_t) - \nabla F(\boldsymbol{\theta}_t)\|^2\right]}_{\mathrm{Err}_2(t)}, \tag{50}$$

where the inequality hold due to $\|\boldsymbol{x} + \boldsymbol{y}\|^2 \leq 2\|\boldsymbol{x}\|^2 + 2\|\boldsymbol{y}\|^2$ for any $\boldsymbol{x}, \boldsymbol{y}$.

Then, we show in the following lemma that, comparing to vanilla sampling-based GCN training, fixing the inner layer sampling inside each recycling epoch will not hurt the feature approximation error.

**Lemma 8** (Lemma 1 in main text). *Denote $\widetilde{D}$ as the number of selected neighbors in the sampling and $d$ as the dimension of node features. With probability at least $1 - \delta$, we have*

$$\frac{1}{2}\mathbb{E}\left[\left\|\nabla \widetilde{F}_{\mathcal{B}}(\boldsymbol{\theta}) - \nabla F_{\mathcal{B}}(\boldsymbol{\theta})\right\|^2\right] \leq 16 L_f^2 G_g^2 \frac{\log(2d/\delta)}{\widetilde{D}} + 32 G_f^2 L_g^4 \frac{\log(2d/\delta) + 1/4}{\widetilde{D}}. \tag{51}$$

The following lemma shows that sampling mini-batches $\mathcal{V}_{\mathcal{B}}(t)$ from a subset of nodes $\mathcal{V}_S(k)$ instead of all nodes will not result in a large gradient approximation error if the size of $\mathcal{V}_S(k)$ is large enough.

**Lemma 9** (Lemma 2 in main text). *Consider the setting where we sample a mini-batch of size $B$ from a subset of $S$ nodes uniformly at random instead of sampling from $N$ nodes. With probability at least $1 - \delta$ we have*

$$\frac{1}{2}\mathbb{E}\left[\|\nabla F_{\mathcal{B}}(\boldsymbol{\theta}) - \nabla F(\boldsymbol{\theta})\|^2\right] \leq 16 G_f^2 \frac{\log(2d/\delta) + 1/4}{B} + 16 G_f^2 \frac{\log(2d/\delta) + 1/4}{S}. \tag{52}$$

Plugging the results in Lemma 8 and Lemma 9 into Eq. 50 yields:

$$\Delta = \mathcal{O}\left(L_f^2 G_g^2 \frac{\log(2d/\delta)}{\widetilde{D}}\right) + \mathcal{O}\left(G_f^2 L_g^4 \frac{\log(2d/\delta) + 1/4}{\widetilde{D}}\right)$$

$$+ \mathcal{O}\left(G_f^2 \frac{\log(2d/\delta) + 1/4}{B}\right) + \mathcal{O}\left(G_f^2 \frac{\log(2d/\delta) + 1/4}{S}\right), \tag{53}$$

where we use $\mathcal{O}(\cdot)$ to hide the constants. By Lemma 3 we know that

$$L_F = L_f G_g + L_g^2 G_f. \tag{54}$$

By plugging the Eq. 53 and Eq. 54 back to Lemma 7, we conclude the proof.

## D.1 Proof of Lemma 7

Recall from the proof of Lemma 4 that we denote stochastic gradient as $\boldsymbol{g}_t := \nabla \widetilde{F}_{\mathcal{B}}(\boldsymbol{\theta})$ and full gradient as $\nabla F(\boldsymbol{\theta})$. From Eq. 32 in the proof of Lemma 4, we know that

$$
\left(\frac{3}{2}\eta - \eta^2 L_F\right) \mathbb{E}[\|\nabla F(\boldsymbol{\theta}_t)\|^2] \leq \mathbb{E}[F(\boldsymbol{\theta}_t)] - \mathbb{E}[F(\boldsymbol{\theta}_{t+1})] + \left(\eta^2 L_F - \frac{1}{2}\eta\right) \mathbb{E}[\|\boldsymbol{g}_t - \nabla F(\boldsymbol{\theta}_t)\|^2]
$$

$$
\leq \mathbb{E}[F(\boldsymbol{\theta}_t)] - \mathbb{E}[F(\boldsymbol{\theta}_{t+1})] + \eta^2 L_F \mathbb{E}[\|\boldsymbol{g}_t - \nabla F(\boldsymbol{\theta}_t)\|^2].
$$

$$(55)$$

Recall that our algorithm runs $K$ recycling epochs $\mathcal{T}_k$ for $k = 1$ to $K$ and the size of each recycling epoch is $|\mathcal{T}_k| = \rho^k R$. Let $\Delta_k = \mathbb{E}[\|\boldsymbol{g}_t - \nabla F(\boldsymbol{\theta}_t)\|^2]$ for any $t \in \mathcal{T}_k$. Then, by summing up above inequality for all $T$ iterations, we have

$$
\left(\frac{3}{2}\eta - \eta^2 L_F\right) \sum_{t=1}^{T} \mathbb{E}[\|\nabla F(\boldsymbol{\theta}_t)\|^2] \leq \mathbb{E}[F(\boldsymbol{\theta}_1)] - \mathbb{E}[F(\boldsymbol{\theta}_\star)] + \eta^2 L_F \sum_{k=1}^{K} \rho^k R \Delta_k. \quad (56)
$$

Dividing both side by $T\left(\frac{3}{2}\eta - \eta^2 L_F\right)$ results in

$$
\frac{1}{T}\sum_{t=1}^{T} \mathbb{E}[\|\nabla F(\boldsymbol{\theta}_t)\|^2] \leq \left(\frac{3}{2}\eta - \eta^2 L_F\right)^{-1} \frac{(\mathbb{E}[F(\boldsymbol{\theta}_1)] - \mathbb{E}[F(\boldsymbol{\theta}_\star)])}{T} + \frac{2\eta^2 L_F \sum_{k=1}^{K} \rho^k R \Delta_k}{(3\eta - 2\eta^2 L_F) T}
$$

$$
= \left(3\eta - 2\eta^2 L_F\right)^{-1} \frac{2\left(\mathbb{E}[F(\boldsymbol{\theta}_1)] - \mathbb{E}[F(\boldsymbol{\theta}_\star)]\right)}{T} + \frac{2\eta L_F \sum_{k=1}^{K} \rho^k R \Delta_k}{(3 - 2\eta L_F) T}.
$$

$$(57)$$

By choosing $\eta = \min\left\{\frac{3}{2L_F}, \sqrt{\frac{\mathbb{E}[F(\boldsymbol{\theta}_1)] - \mathbb{E}[F(\boldsymbol{\theta}_\star)]}{L_F(\sum_{k=1}^{K} \rho^k R \Delta_k)}}\right\}$ and noting that $\tilde{\boldsymbol{\theta}}$ is decided in a way that has minimum gradient we have

$$
\mathbb{E}[\|\nabla F(\tilde{\boldsymbol{\theta}})\|^2] \leq \frac{1}{T}\sum_{t=1}^{T} \mathbb{E}[\|\nabla F(\boldsymbol{\theta}_t)\|^2]
$$

$$
\leq \frac{1}{\eta} \frac{2\left(\mathbb{E}[F(\boldsymbol{\theta}_1)] - \mathbb{E}[F(\boldsymbol{\theta}_\star)]\right)}{T} + \frac{2\eta L_F \sum_{k=1}^{K} \rho^k R \Delta_k}{T} \quad (58)
$$

$$
\leq \mathcal{O}\left(\frac{\sqrt{\sum_{k=1}^{K} \rho^k R \Delta_k}}{T}\right),
$$

which gives the bound as stated.

**Remark 1.** *Consider a total budget of $T$ fresh mini-batch samplings for vanilla GCNs. When we use fixed size recycling stages, i.e., $\rho = 1$, the obtained rate for* LAZYGCN *reduces to $\sqrt{\Delta_{\text{LAZYGCN}}/T}$, the same as convergence rate of SGCN, but with a difference in the mean-square error of stochastic gradients, i.e., $\Delta_{\text{LAZYGCN}}$ versus $\Delta_{SGCN}$. For $\rho > 1$, the size of recycling stages will gradually increases, resulting in a decrease in the number of samplings $K$ required, which in turn shortens the total training time. Besides, as the training progresses, the mean-square error of stochastic gradient will gradually decrease as the gradient decreases. Therefore, a careful gradual increase in the recycling stage size will not cause a sharp increase of $\Delta_{\text{LAZYGCN}}$.*

## D.2 Proof of Lemma 8

By definition, we know $\nabla F_{\mathcal{B}}(\boldsymbol{\theta})$ and $\nabla \widetilde{F}_{\mathcal{B}}(\boldsymbol{\theta})$ are defined as

$$
\nabla F_{\mathcal{B}}(\boldsymbol{\theta}) = \frac{1}{B}\sum_{i \in \mathcal{V}_{\mathcal{B}}} \nabla f_i \underbrace{\left(\frac{1}{D}\sum_{j \in \mathcal{N}(i)} g_j(\boldsymbol{\theta})\right)}_{A_1(i)} \underbrace{\left(\frac{1}{D}\sum_{j \in \mathcal{N}(i)} \nabla g_j(\boldsymbol{\theta})\right)}_{A_2(i)}, \quad (59)
$$

$$\nabla \widetilde{F}_{\mathcal{B}}(\boldsymbol{\theta}) = \frac{1}{B} \sum_{i \in \mathcal{V}_{\mathcal{B}}} \underbrace{\nabla f_i \left( \frac{1}{\widetilde{D}} \sum_{j \in \widetilde{\mathcal{N}}(i)} g_j(\boldsymbol{\theta}) \right)}_{B_1(i)} \underbrace{\left( \frac{1}{\widetilde{D}} \sum_{j \in \widetilde{\mathcal{N}}(i)} \nabla g_j(\boldsymbol{\theta}) \right)}_{B_2(i)}. \tag{60}$$

For simplicity, we denote $\mathbb{E}_{\mathcal{V}_{\mathcal{S}} \sim \mathcal{V}} \left[ \mathbb{E}_{j \sim \mathcal{N}(i), \forall i \in \mathcal{V}_{\mathcal{B}} \mid \mathcal{V}_{\mathcal{B}} \sim \mathcal{V}_{\mathcal{S}}} [\cdot] \right]$ as $\mathbb{E}[\cdot]$.

$$\begin{aligned}
& \mathbb{E} \left[ \left\| \nabla \widetilde{F}_{\mathcal{B}}(\boldsymbol{\theta}) - \nabla F_{\mathcal{B}}(\boldsymbol{\theta}) \right\|^2 \right] \\
&= \mathbb{E} \left[ \left\| \frac{1}{B} \sum_{i \in \mathcal{V}_{\mathcal{B}}} B_1(i) B_2(i) - \frac{1}{B} \sum_{i \in \mathcal{V}_{\mathcal{B}}} A_1(i) A_2(i) \right\|^2 \right] \\
&\underset{(a)}{\leq} \mathbb{E} \left[ \| B_1(i) B_2(i) - A_1(i) A_2(i) \|^2 \right] \\
&\underset{(b)}{\leq} 2\mathbb{E} \left[ \| B_1(i)(B_2(i) - A_2(i)) \|^2 \right] + 2\mathbb{E} \left[ \| (B_1(i) - A_1(i)) A_2(i) \|^2 \right] \\
&\underset{(c)}{\leq} 2\mathbb{E} \left[ \| B_1(i) \|^2 \right] \mathbb{E} \left[ \| B_2(i) - A_2(i) \|^2 \right] + 2\mathbb{E} \left[ \| B_1(i) - A_1(i) \|^2 \right] \mathbb{E} \left[ \| A_2(i) \|^2 \right],
\end{aligned} \tag{61}$$

where inequality (a) is due to $\| \frac{1}{n} \sum_{i=1}^n \boldsymbol{x}_i \| \leq \frac{1}{n} \sum_{i=1}^n \| \boldsymbol{x}_i \|$, inequality (b) is due to $\| \boldsymbol{x} + \boldsymbol{y} \|^2 \leq 2\| \boldsymbol{x} \| + 2\| \boldsymbol{y} \|$, and inequality (c) is due to $\| \boldsymbol{xy} \| \leq \| \boldsymbol{x} \| \| \boldsymbol{y} \|$.

(1) Considering $\mathbb{E} \left[ \| B_1(i) \|^2 \right]$, we have

$$\mathbb{E} \left[ \| B_1(i) \|^2 \right] = \mathbb{E} \left[ \left\| \nabla f_i \left( \frac{1}{\widetilde{D}} \sum_{j \in \widetilde{\mathcal{N}}(i)} g_j(\boldsymbol{\theta}) \right) \right\|^2 \right] \leq L_f^2. \tag{62}$$

(2) Considering $\mathbb{E} \left[ \| A_2(i) \|^2 \right]$, we have

$$\mathbb{E} \left[ \| A_2(i) \|^2 \right] = \mathbb{E} \left[ \left\| \frac{1}{D} \sum_{j \in \mathcal{N}(i)} \nabla g_j(\boldsymbol{\theta}) \right\|^2 \right] \leq L_g^2. \tag{63}$$

(3) Considering $\mathbb{E} \left[ \| B_1(i) - A_1(i) \|^2 \right]$, we have

$$\begin{aligned}
\mathbb{E} \left[ \| B_1(i) - A_1(i) \|^2 \right] &= \mathbb{E} \left[ \left\| \nabla f_i \left( \frac{1}{\widetilde{D}} \sum_{j \in \widetilde{\mathcal{N}}(i)} g_j(\boldsymbol{\theta}) \right) - \nabla f_i \left( \frac{1}{D} \sum_{j \in \mathcal{N}(i)} g_j(\boldsymbol{\theta}) \right) \right\|^2 \right] \\
&\leq G_f^2 \mathbb{E} \left[ \left\| \frac{1}{\widetilde{D}} \sum_{j \in \widetilde{\mathcal{N}}(i)} g_j(\boldsymbol{\theta}) - \frac{1}{D} \sum_{j \in \mathcal{N}(i)} g_j(\boldsymbol{\theta}) \right\|^2 \right] \\
&\leq 32 G_f^2 L_g^2 \frac{\log(2d/\delta) + 1/4}{\widetilde{D}},
\end{aligned} \tag{64}$$

where the last inequality is due to Lemma 12.

(4) Considering $\mathbb{E} \left[ \| B_2(i) - A_2(i) \|^2 \right]$, we have

$$\begin{aligned}
\mathbb{E} \left[ \| B_2(i) - A_2(i) \|^2 \right] &= \mathbb{E} \left[ \left\| \frac{1}{\widetilde{D}} \sum_{j \in \widetilde{\mathcal{N}}(i)} \nabla g_j(\boldsymbol{\theta}) - \frac{1}{D} \sum_{j \in \mathcal{N}(i)} \nabla g_j(\boldsymbol{\theta}) \right\|^2 \right] \\
&\leq 16 G_g^2 \frac{\log(2d/\delta)}{\widetilde{D}},
\end{aligned} \tag{65}$$

where the last inequality is due to Lemma 12.

(6) By plugging Eq. 62, Eq. 63, Eq. 64, and Eq. 65 back to Eq. 61, it holds that with probability at least $1 - \delta$

$$\mathbb{E}\left[\left\|\nabla\widetilde{F}_{\mathcal{B}}(\boldsymbol{\theta}) - \nabla F_{\mathcal{B}}(\boldsymbol{\theta})\right\|^2\right] \le 32L_f^2 G_g^2 \frac{\log(2d/\delta)}{\widetilde{D}} + 64G_f^2 L_g^4 \frac{\log(2d/\delta) + 1/4}{\widetilde{D}}. \tag{66}$$

### D.3 Proof of Lemma 9

We define the objective function for sampling a mini-batch of size $B$ from a subset of $S$ nodes uniformly at random instead of sampling from $N$ nodes as $F_{\mathcal{S}}$ and its gradient as $\nabla F_{\mathcal{S}}(\boldsymbol{\theta})$. Now we have

$$\nabla F_{\mathcal{B}}(\boldsymbol{\theta}) = \frac{1}{B}\sum_{i\in\mathcal{V}_{\mathcal{B}}} \nabla f_i \underbrace{\left(\frac{1}{D}\sum_{j\in\mathcal{N}(i)} g_j(\boldsymbol{\theta})\right)}_{A_1(i)} \underbrace{\left(\frac{1}{D}\sum_{j\in\mathcal{N}(i)} \nabla g_j(\boldsymbol{\theta})\right)}_{A_2(i)}, \tag{67}$$

$$\nabla F_{\mathcal{S}}(\boldsymbol{\theta}) = \frac{1}{S}\sum_{i\in\mathcal{V}_{\mathcal{S}}} \nabla f_i \underbrace{\left(\frac{1}{D}\sum_{j\in\mathcal{N}(i)} g_j(\boldsymbol{\theta})\right)}_{A_1(i)} \underbrace{\left(\frac{1}{D}\sum_{j\in\mathcal{N}(i)} \nabla g_j(\boldsymbol{\theta})\right)}_{A_2(i)}, \tag{68}$$

$$\nabla F(\boldsymbol{\theta}) = \frac{1}{N}\sum_{i\in\mathcal{V}} \nabla f_i \underbrace{\left(\frac{1}{D}\sum_{j\in\mathcal{N}(i)} g_j(\boldsymbol{\theta})\right)}_{A_1(i)} \underbrace{\left(\frac{1}{D}\sum_{j\in\mathcal{N}(i)} \nabla g_j(\boldsymbol{\theta})\right)}_{A_2(i)}. \tag{69}$$

For simplicity, we denote $\mathbb{E}_{\mathcal{V}_{\mathcal{S}}\sim\mathcal{V}}\left[\mathbb{E}_{j\sim\mathcal{N}(i),\forall i\in\mathcal{V}_{\mathcal{B}} \mid \mathcal{V}_{\mathcal{B}}\sim\mathcal{V}_{\mathcal{S}}}[\cdot]\right]$ as $\mathbb{E}[\cdot]$.

Then, we have

$$\begin{aligned}
\mathbb{E}\left[\|\nabla F_{\mathcal{B}}(\boldsymbol{\theta}) - \nabla F(\boldsymbol{\theta})\|^2\right] &= \mathbb{E}\left[\|\nabla F_{\mathcal{B}}(\boldsymbol{\theta}) - \nabla F_{\mathcal{S}}(\boldsymbol{\theta})\|^2\right] + \mathbb{E}\left[\|\nabla F_{\mathcal{S}}(\boldsymbol{\theta}) - \nabla F(\boldsymbol{\theta})\|^2\right] \\
&\quad + 2\mathbb{E}\left[\langle\nabla F_{\mathcal{B}}(\boldsymbol{\theta}) - \nabla F_{\mathcal{S}}(\boldsymbol{\theta}), \nabla F_{\mathcal{S}}(\boldsymbol{\theta}) - \nabla F(\boldsymbol{\theta})\rangle\right] \\
&\overset{(a)}{=} \mathbb{E}\left[\left\|\frac{1}{B}\sum_{i\in\mathcal{V}_{\mathcal{B}}} A_1(i)A_2(i) - \frac{1}{S}\sum_{i\in\mathcal{V}_{\mathcal{S}}} A_1(i)A_2(i)\right\|^2\right] \\
&\quad + \mathbb{E}\left[\left\|\frac{1}{S}\sum_{i\in\mathcal{V}_{\mathcal{S}}} A_1(i)A_2(i) - \frac{1}{N}\sum_{i\in\mathcal{V}} A_1(i)A_2(i)\right\|^2\right] \\
&\overset{(b)}{\le} 32G_f^2 \frac{\log(2d/\delta) + 1/4}{B} + 32G_f^2 \frac{\log(2d/\delta) + 1/4}{S},
\end{aligned} \tag{70}$$

where the equality (a) is due to $\mathbb{E}\left[\nabla F_{\mathcal{B}}(\boldsymbol{\theta}) - \nabla F_{\mathcal{S}}(\boldsymbol{\theta})\right] = \mathbb{E}\left[\nabla F_{\mathcal{S}}(\boldsymbol{\theta}) - \nabla F(\boldsymbol{\theta})\right] = 0$ and inequality (b) is due to Lemma 12.

## E  Useful Theorems and Lemmas

In this section we present a host of lemmas that are used in presentation of the proofs. We start by the following results that exhibit the sums of independent random vectors and matrices provide normal concentration near its mean in a range determined by the variance of the sum.

**Theorem 3** (Vector Bernstein Inequality [12]). *Let $\boldsymbol{x}_1, .., \boldsymbol{x}_n$ be independent random vectors with common dimension $d$ and assume that each one is centered, uniformly bounded, and the variance is bounded, i.e.,*

$$\mathbb{E}[\boldsymbol{x}_j] = 0 \text{ and } \|\boldsymbol{x}_j\| \le \mu \text{ as well as } \|\mathbb{E}[\boldsymbol{x}_j^2]\| \le \sigma^2. \tag{71}$$

*Let introduce the sum of $n$ vectors as*

$$\boldsymbol{z} = \frac{1}{n}\sum_{j=1}^{n}\boldsymbol{x}_j, \tag{72}$$

*then we have*

$$\mathbb{P}(\|\boldsymbol{z}\| \geq \epsilon) \leq 2d \cdot \exp\left(-n \cdot \frac{\epsilon^2}{8\sigma^2} + \frac{1}{4}\right). \tag{73}$$

**Theorem 4** (Matrix Bernstein Inequality [12])**.** *Let $\boldsymbol{X}_1, .., \boldsymbol{X}_n$ be independent random matrices with common dimension $d \times d$ and assume that each one is centered, uniformly bounded, and the variance is bounded, i.e.,*

$$\mathbb{E}[\boldsymbol{X}_j] = 0 \text{ and } \|\boldsymbol{X}_j\| \leq \mu \text{ as well as } \|\mathbb{E}[\boldsymbol{X}_j^2]\| \leq \sigma^2. \tag{74}$$

*Let introduce the sum of $n$ matrices as*

$$\mathbf{Z} = \frac{1}{n}\sum_{j=1}^{n}\boldsymbol{X}_j, \tag{75}$$

*then we have*

$$\mathbb{P}(\|\mathbf{Z}\| \geq \epsilon) \leq 2d \cdot \exp\left(-n \cdot \min\left\{\frac{\epsilon^2}{4\sigma^2}, \frac{\epsilon}{2\mu}\right\}\right). \tag{76}$$

The following lemma arises from the vector Bernstein inequality in Theorem 3, which consistutes a non-asymptotic bound on the function that has a vector as output that holds with high probability.

**Lemma 10** (Lemma 6 in [16])**.** *Let the sub-sampled function defined as*

$$V_{\mathcal{S}}(\boldsymbol{\theta}) = \frac{1}{S}\sum_{i \in \mathcal{S}} V_j(\boldsymbol{\theta}), \ \mathbb{E}\left[V_{\mathcal{S}}(\boldsymbol{\theta})\right] = V(\boldsymbol{\theta}), \tag{77}$$

*where $V_j(\boldsymbol{\theta}) \in \mathbb{R}^d$ is $\rho$-Lipschitz continuous for all $j$. For $\epsilon \leq 2\rho$, we have with probability at least $1 - \delta$ that*

$$\|V_{\mathcal{S}}(\boldsymbol{\theta}) - V(\boldsymbol{\theta})\| \leq 4\sqrt{2}\rho\sqrt{\frac{\log(2d/\delta) + 1/4}{S}}. \tag{78}$$

The following lemma arises from the matrix Bernstein inequality in Theorem 4, which consistutes a non-asymptotic bound on the function that has a matrix as output that holds with high probability.

**Lemma 11** (Lemma 8 in [16])**.** *Let the sub-sampled function defined as*

$$M_{\mathcal{S}}(\boldsymbol{\theta}) = \frac{1}{S}\sum_{j \in \mathcal{S}} M_j(\boldsymbol{\theta}), \ \mathbb{E}\left[M_{\mathcal{S}}(\boldsymbol{\theta})\right] = M(\boldsymbol{\theta}), \tag{79}$$

*where $M_j(\boldsymbol{\theta}) \in \mathbb{R}^{d \times d}$ is $\rho$-Lipschitz continuous for all $j$. For $\epsilon \leq 4\rho$, we have with probability at least $1 - \delta$ that*

$$\|M_{\mathcal{S}}(\boldsymbol{\theta}) - M(\boldsymbol{\theta})\| \leq 4\rho\sqrt{\frac{\log(2d/\delta)}{S}}. \tag{80}$$

The following lemma provides a high probability upper-bound using vector concentration inequality in Lemma 10 and matrix concentration inequality in Lemma 11 on the function approximation variance and gradient approximation variance, respectively.

**Lemma 12.** *Consider the setting where we sample a mini-batch of size $B$ from a subset of $S$ nodes uniformly at random instead of sampling from $N$ nodes. Let $\tilde{D}$ as the number of the selected*

*neighbors in the sampling. With probability at least $1 - \delta$ we have*

$$
\left\| \frac{1}{\widetilde{D}} \sum_{j \in \widetilde{\mathcal{N}}(i)} g_j(\boldsymbol{\theta}) - \mathbb{E}_{j \sim \mathcal{N}(i)}\left[g_j(\boldsymbol{\theta})\right] \right\| \underset{(a)}{\leq} 4\sqrt{2}L_g \sqrt{\frac{\log(2d/\delta) + 1/4}{\widetilde{D}}}, \text{ for any } i \in \mathcal{V},
$$

$$
\left\| \frac{1}{\widetilde{D}} \sum_{j \in \widetilde{\mathcal{N}}(i)} \nabla g_j(\boldsymbol{\theta}) - \mathbb{E}_{j \sim \mathcal{N}(i)}\left[\nabla g_j(\boldsymbol{\theta})\right] \right\| \underset{(b)}{\leq} 4G_g \sqrt{\frac{\log(2d/\delta)}{\widetilde{D}}}, \text{ for any } i \in \mathcal{V},
$$

$$
\left\| \frac{1}{B} \sum_{i \in \mathcal{V}_{\mathcal{B}}} \nabla f_i(g(\boldsymbol{\theta})) - \mathbb{E}_{i \sim \mathcal{V}_{\mathcal{S}}}\left[\nabla f_i(g(\boldsymbol{\theta}))\right] \right\| \underset{(c)}{\leq} 4\sqrt{2}G_f \sqrt{\frac{\log(2d/\delta) + 1/4}{B}}, \text{ for any } \mathcal{V}_{\mathcal{B}} \subseteq \mathcal{V}_{\mathcal{S}},
$$

$$
\left\| \frac{1}{S} \sum_{i \in \mathcal{V}_{\mathcal{S}}} \nabla f_i(g(\boldsymbol{\theta})) - \mathbb{E}_{i \in \mathcal{V}}\left[\nabla f_i(g(\boldsymbol{\theta}))\right] \right\| \underset{(d)}{\leq} 4\sqrt{2}G_f \sqrt{\frac{\log(2d/\delta) + 1/4}{S}}, \text{ for any } \mathcal{V}_{\mathcal{S}} \subseteq \mathcal{V}.
$$
(81)

### E.1 Proof of Lemma 10

The proof is generalized from the proof of Lemma 6 in [16]. Since the expectation of the random matrices need to be zero, we first center the individual matrices

$$
X_j = V_j(\boldsymbol{\theta}) - V(\boldsymbol{\theta}), \ \forall i \in \mathcal{S}.
$$
(82)

Since $V_j(\boldsymbol{\theta})$ is $\rho$-Lipschitz continuous, we have for all $i \in \mathcal{S}$,

$$
\|X_j\| = \|V_j(\boldsymbol{\theta}) - V(\boldsymbol{\theta})\| \leq 2\rho,
$$
(83)

and

$$
\|X_j\|^2 = \|V_j(\boldsymbol{\theta}) - V(\boldsymbol{\theta})\|^2 \leq 4\rho^2.
$$
(84)

Therefore, for any $\epsilon \leq 2\rho$, we are in the small deviation regime of Bernstein's bound with a sub-Gaussian tail. Then, by using Theorem 3 and plugging back

$$
\frac{1}{S} \sum_{i \in \mathcal{S}} V_j(\boldsymbol{\theta}) - V(\boldsymbol{\theta}),
$$
(85)

we have

$$
\mathbb{P}\left( \left\| \frac{1}{S} \sum_{j \in \mathcal{S}} V_j(\boldsymbol{\theta}) - V(\boldsymbol{\theta}) \right\| \geq \epsilon \right) \leq 2d \cdot \exp\left( -\frac{\epsilon^2 B}{32\rho^2} + \frac{1}{4} \right).
$$
(86)

Finally, let $\delta$ as the upper bound of Bernstein inequality

$$
\delta = 2d \cdot \exp\left( -\frac{\epsilon^2 B}{32\rho^2} \right).
$$
(87)

Therefore, we have

$$
\epsilon = 4\sqrt{2}\rho \sqrt{\frac{\log(2d/\delta) + 1/4}{S}}.
$$
(88)

### E.2 Proof of Lemma 11

The proof is generalized from the proof of Lemma 8 in [16]. Since the expectation of the random matrices need to be zero, we first center the individual matrices

$$
\boldsymbol{X}_j = M_j(\boldsymbol{\theta}) - M(\boldsymbol{\theta}), \ \forall i \in \mathcal{S}.
$$
(89)

Since $M_j(\boldsymbol{\theta})$ is $\rho$-Lipschitz continuous, we have for all $i \in \mathcal{S}$,

$$
\|\boldsymbol{X}_j\| = \|M_j(\boldsymbol{\theta}) - M(\boldsymbol{\theta})\| \leq 2\rho,
$$
(90)

and

$$\|\boldsymbol{X}_j\|^2 = \|M_j(\boldsymbol{\theta}) - M(\boldsymbol{\theta})\|^2 \leq 4\rho^2. \tag{91}$$

Therefore, for any $\epsilon \leq 4\rho$, we are in the small deviation regime of Bernstein's bound with a sub-Gaussian tail. Then, by using Theorem 4 and plugging back

$$\frac{1}{S}\sum_{j \in \mathcal{S}} M_j(\boldsymbol{\theta}) - M(\boldsymbol{\theta}), \tag{92}$$

we have

$$\mathbb{P}\left(\left\|\frac{1}{S}\sum_{j \in \mathcal{S}} M_j(\boldsymbol{\theta}) - M(\boldsymbol{\theta})\right\| \geq \epsilon\right) \leq 2d \cdot \exp\left(-\frac{\epsilon^2 B}{16\rho^2}\right). \tag{93}$$

Finally, let $\delta$ as the upper bound of Bernstein inequality

$$\delta = 2d \cdot \exp\left(-\frac{\epsilon^2 B}{16\rho^2}\right). \tag{94}$$

Therefore, we have

$$\epsilon = 4\rho\sqrt{\frac{\log(2d/\delta)}{S}}. \tag{95}$$

### E.3 Proof of Lemma 12

The proof of inequality (a) follows from Lemma 10. Specifically, let $\mathcal{S} = \widetilde{\mathcal{N}}(i)$ of size $S = \widetilde{D}$, and let $V_j(\boldsymbol{\theta}) = g_j(\boldsymbol{\theta})$ for any $j \in \widetilde{\mathcal{N}}(i)$. By the Lipschitz continuous assumption, we know that $V_j(\boldsymbol{\theta}) \in \mathbb{R}^d$ is $L_g$-Lipschitz continuous. Therefore, by Lemma 10, we have

$$\|V_{\mathcal{S}}(\boldsymbol{\theta}) - V(\boldsymbol{\theta})\| = \left\|\frac{1}{\widetilde{D}}\sum_{j \in \widetilde{\mathcal{N}}(i)} g_j(\boldsymbol{\theta}) - \mathbb{E}_{j \sim \mathcal{N}(i)}\left[g_j(\boldsymbol{\theta})\right]\right\| \leq 4\sqrt{2}L_g\sqrt{\frac{\log(2d/\delta) + 1/4}{\widetilde{D}}}. \tag{96}$$

For (b), noting that $\nabla g_j(\boldsymbol{\theta})$ is a matrix with dimension $d \times d$, we can apply matrix Bernstein inequality in Lemma 11 to bound the deviation of averaged matrices over $\tilde{D}$ from its expected value. Let $\mathcal{S} = \widetilde{\mathcal{N}}(i)$ of size $S = \widetilde{D}$, and let $M_j(\boldsymbol{\theta}) = \nabla g_j(\boldsymbol{\theta})$ for any $j \in \widetilde{\mathcal{N}}(i)$. By the Lipschitz continuous assumption, we know that $M_j(\boldsymbol{\theta}) \in \mathbb{R}^{d \times d}$ is $G_g$-Lipschitz continuous. Therefore, we have

$$\|M_{\mathcal{S}}(\boldsymbol{\theta}) - M(\boldsymbol{\theta})\| = \left\|\frac{1}{\widetilde{D}}\sum_{j \in \widetilde{\mathcal{N}}(i)} \nabla g_j(\boldsymbol{\theta}) - \mathbb{E}_{j \sim \mathcal{N}(i)}\left[\nabla g_j(\boldsymbol{\theta})\right]\right\| \leq 4G_g\sqrt{\frac{\log(2d/\delta)}{\widetilde{D}}}. \tag{97}$$

The inequalities in (c) and (d) follow similarly.