[Reviews · NeurIPS 2020]

Review 1

Summary and Contributions: The paper introduces LazyGCN, a method to optimise CPU/GPU utilisation while training sampling-based GCN models. With a pipeline that introduces periodic sampling and recycling of samples for the inner layers during the training phase, the algorithm obtains results comparable to other state-of-the-art algorithms with a fraction of the training time.

Strengths: The paper introduces a new pipeline to training sampling-based GCN models in a fraction of the time with comparable performances. This is mainly a practical result, however motivated by a theoretical analysis which looks sound to me. I found the method interesting and I think it could spark conversation about how to better buffer / recycle information coming from graphs. Tackling this problem in an efficient and scalable way is definitely relevant at the moment.

Weaknesses: I consider the advantage of using this method somehow limited, as it brings significant improvements at training time only, while other approaches (such as e.g. SIGN) aim at optimisations which allow for faster inference too, and without recurring to sampling. As a minor note, the paper emphasises sampling time as a huge bottleneck in the training pipeline, however I’d argue that this is not the main issue (as samples can be prepared on more CPUs, buffered into a queue in the main RAM, and simply served when required - still costly but in a negligible way compared to the GPU resources). This is somewhat shown in the VanillaGCN step of Figure1. The real issue is, in my opinion, the bottleneck introduced by transferring data to the GPU, which LazyGCN is able to properly address.

Correctness: Claims and methods look correct to me. The proposed methodology is evaluated wrt the three main families of sampling tasks, comparing with their respective SOA methods.

Clarity: The paper is well written overall, however there are few typos that should be corrected (e.g. T/R as a superscript at line 156, “convergence” at line 188, “mini bath” at 129, etc).

Relation to Prior Work: The work cites previous works on sampling approaches for GCNs but does not refer to comparable methods for optimising training time / convergence (actually I am not aware of other methods for GCNs specifically tackling the training pipeline and the optimisation of the communication between CPU and GPU)

Reproducibility: Yes

Additional Feedback: The authors properly addressed my (already few) concerns so I have updated my score. Thanks!


Review 2

Summary and Contributions: To enable more hardware-friendly GCN computation, this paper proposes a disentanglement of sampling and computation. In detail, CPU that conductes data sampling and transfer applies periodic sampling to reduce time cost, and GPU that performs GCN computation recycles the already-sampled nodes for a predetermined number of iterations. Theoretical analysis is derived to show that the proposed LazyGCN enjoys the same convergence rate as the vanilla sampling methods, with reduction of CPU sampling. Experiments are conducted on several real graphs, where the claims by this paper are generally supported.

Strengths: 1. Generally well-written paper. 2. The idea of recycling sampling with fixing inner layer nodes is interesting and novel. The weakness of current sampling-based GCNs mostly comes from the cost of sampling and data transfer between CPU and GPU. So the solution by this paper is valuable and well motivated. 3. The theorems are good and hit the point, although there still some questions that required to address. 4. Experimental evluations are convinced and almost sufficient.

Weaknesses: There are several concerns that prevent me from raising my score for the current version. 1. The statements around Line 161-163 seems a bit arbitrary at first. Does any theoretical evidence or experimental observation that can support why this happens? 2. Line 189-190 is problemetic. GCN has multi-level composite structure. That means the variances of different layers are influenced with each other. It is not straightforward to extend the analysis of single-layer GCN to its multi-layer version. 3. According to Eq. (10), it seems larger \rho will lead to larger bound. So why we need \rho>1?

Correctness: Generally correct. Some suggestions are provided to make this paper more solid. 1. The datasets applied in this paper are still not so big. Perhaps, trying a larger scale dataset (such as Amazon in GraphSAINT) will be more desirable to support the benefit of the proposed method. 2. What does "epoch" of LazyGCN mean in Figure 2. (b)? Is it equal to RK? And, it is better to compare the time per epoch between the methods with and without Lazy sampling. 3. Something is wrong with Figure 2 (c). The total time of +LG should be much smaller than those methods without LG.

Clarity: This paper is generally well written. Yet some issues are required to be addressed, see above. Eq. (7): \delta F_B -> \delta F ?

Relation to Prior Work: The citations of sampliny-based GCNs are insufficient. The node-wise method [A] and layer-wise method [B] are missing and should be included. Both [20] and [B] resort to sampling on CPU and computation on GPU, which makes them desirable baselines to demonstrate the performance of the proposed recycling sampling. It is suggested to include the comparison with them in the experiment. [A] tochastic training of graph convolutional networks with variance reduction. ICML 2018. [B] Adaptive sampling towards fast graph representation learning. NIPS, 2018.

Reproducibility: Yes

Additional Feedback: I am open to listen to the authors' feedback and will change my score if all concerns are addressed. ### post rebuttal ### All concerns are addressed; accept.


Review 3

Summary and Contributions: GCNs are typically trained in CPU-GPU heterogeneous systems since sampling is inefficient on GPUs. Yet this design incurs low efficiency due to bandwidth and memory bottlenecks. This submission suggests decoupling the frequency of sampling from the sampling strategy (i.e., to periodically perform sampling and allow the samples to be reused, instead of insisting on immediately performing sampling every time a batch is consumed). It gives extensive theoretical analyses on the convergence of the proposed strategy.

Strengths: 1. It identifies a key efficiency issue with GNNs and proposes an easy-to-implement solution. 2. Extensive theoretical analyses on the convergence of the proposed solution. 3. Relatively comprehensive empirical evaluation. 4. The proposed method may be even more relevant in a distributed training environment, where one might use a few workers to perform sampling and transfer the samples to the workers that are responsible for performing computation on GPUs.

Weaknesses: The datasets are of small or medium scale. It lacks large-scale experiments (ideally in a distributed training environment) to demonstrate the merits of the proposed method.

Correctness: Yes.

Clarity: Yes.

Relation to Prior Work: Yes.

Reproducibility: Yes

Additional Feedback:


Review 4

Summary and Contributions: To reduce the computing time of GCN spending on preprocessing and loading new samples, this paper proposed LazyGCN to perform sampling periodically and effectively recycled the sampled nodes to mitigate data preparation overhead. The convergence of LazyGCN was theoretically analyzed. The effectiveness of LazyGCN was also shown in the experimental evaluation results.

Strengths: The paper is very well written. The idea of recycling the already-sampled nodes on GPU is a simple idea. But it works well on reducing the time for sampling nodes to feed for node embedding update. The proposed LazyGCN uses much less training time than the Vanilla GCN, while the performance on node classification is similar.

Weaknesses: The largest evaluation dataset is Yelp, which has 716K nodes. To demonstrate the capability of LazyGCN on reducing the training time of GCN, it would be more convincing if there is an evaluation on a larger graph than Yelp. Scalability should be analyzed by showing how the training increases with the increase of graph size.

Correctness: The evaluation results show the superior performance of the proposed solution. In addition, extensive results in the supplementary document further confirms the effectiveness of the proposed LazyGCN.

Clarity: The paper is very well written.

Relation to Prior Work: The related work discussion is sufficient.

Reproducibility: Yes

Additional Feedback: It would be more convincing if there is an evaluation on a larger graph than Yelp. Scalability should be analyzed by showing how the training increases with the increase of graph size. Authors' response addressed my comments.

[Author Response · NeurIPS 2020]

We are grateful for the useful and constructive comments from all anonymous reviewers. **(Large dataset)** As suggested by reviewers, we report the result of `LazyGCN` on Amazon dataset (1.5m nodes) in Table A1.

Table A1: Comparison of the test F1 score and time (addition to Table 2)

|  |  | NodeWise | **+LG** | LayerWise | **+LG** | SubGraph | **+LG** |
|---|---|---|---|---|---|---|---|
| `Amazon` | Testing F1 | 77.29% | 76.99% | 77.23% | 77.10% | 77.25% | 77.12% |
|  | Time (s) | 5092.8 | **463.9** | 571.4 | **206.8** | 385.2 | **198.4** |

**Reviewer 1 (Train vs inference)** We thank the reviewer for the thoughtful comments. In this paper, we aim to reduce training time for sampling-based GCN, which has been widely used to scale GCN training on extremely large graphs. Upon carefully reading of the suggested paper by reviewer, we note that although this paper also proposes a new method to accelerate GCN training and inference, it requires a specially designed GCN structure, while our method can be jointly used with any sampling-based GCN models, such as `GraphSAGE`, `FastGCN`, and `GAT`. **(Sample / transfer)** We appreciate bringing up this matter. It is worth mentioning that sampling time increases significantly as the graph size grows, while the transfer and computation time on GPUs remain the same, given the limited GPU capacity. In fact, there are a few key factors that can degrade the performance of sampling for GCN: (a) Unlike GPU that only process a fraction of data, CPUs need to work on the entire graph, which imposes significant overheads for both computation (random memory accesses for traversing the large data structure) and storage (limited main memory); (b) When the size of the graph is large, the graph may not be fully loaded into the main memory (e.g., RAM), and it must be stored in secondary storage (disk). Doing so will incur extra time to transfer data from secondary storage to main memory. Besides, as shown in Figure 1, the GPU is stalled after mini-batch 2 due to the sampling process of a mini-batch 3. Although in this case, the GPU idle time seems negligible compared to the overhead of transfer, when sampling time increases (for a fixed number of CPUs), the idle time will further slow down the training. We will make sure to highlight this overhead in the revision. **(Related work)** To the best of our knowledge, we are the first paper working on accelerating the GCN training by reducing the sampling and transfer time. Besides, similar problems faced in other fields are discussed in lines 75-78.

**Reviewer 2 (Why $\rho > 1$)** The intuition comes from *exploration* and *exploiting* trade-off in standard convex and non-convex optimization analysis. At the beginning of training, our solution $\boldsymbol{\theta}_t$ is far away from stationary point $\boldsymbol{\theta}^\star$ and the gradient $\|\nabla F(\boldsymbol{\theta}_t)\|$ is large. At this point in time, more recycling on a sampled mini-batch might cause an overfit on that mini-batch, while more fresh samples (less recycling) enable us to find the right direction toward optimal solution (exploring). As the optimization proceed, the gradient vanishes and $\|\nabla F(\boldsymbol{\theta}_t)\|$ becomes small. As a result, the possibility of overfitting is much smaller, which allows for more recycling (exploiting). Besides, as suggested by reviewer, we demonstrate the effect of different $\rho$ on the convergence of LayerWise method in the figure on the right hand side. It can be seen that smaller $\rho$ requires more time to recover from overfitted model. **(Multi-level version)** We thank reviewer for the careful reading. Indeed, the variance at the $\ell$th layer will affect the variance of $\{\ell+1, \ldots, L\}$ layers, where $L$ is the number of layers. We will provide the analysis for the multi-level case in the subsequent version. However, we want to point out that the single layer GCN can be formulated as a two level optimization problem ($L$-layer GCN can be formulated as $L+1$ level optimization problem) where the variance at the 1st level already

influence the variance at the 2nd level. **(Effect of $\rho$ on upper bound)** We note that the total number of iterations is constant, hence larger $\rho$ leads to smaller $K$ and does not affect the bound. **(Epoch size)** Note that the size of the $k$th epoch is $\rho^k R$ iterations in Algorithm 1, which is increasing during training, while the epoch size is fixed in vanilla GCN. We illustrate the validation scores for every 10 iterations for both settings to make the figure readable. **(Figure 2(c) clarification)** In this figure we want to show `LazyGCN` can increase *the fraction of time on computing* during training. The figure is normalized (divided) by total wall-clock time to show the proportion of each phase and the total wall-clock time are reported in Table 2. **(Notation $\mathcal{B}$)** Eq.7 shows the stochastic gradient computed on mini-batch $\mathcal{B}$ in `LazyGCN` is close to the stochastic gradient of GCN without node sampling. Therefore, the subscript $\mathcal{B}$ shouldn't be ignored. **(Related works)** We thank reviewer for the advise. Our goal is to develop a general yet effective framework that can be integrated with any sampling strategy to substantially improve the training time. Our experiments are conducted under the same setups (e.g., GCN architecture and hyper-parameters) as the backbone method. Notice that both [A] and [B] can be regarded as an incremental method of `GraphSAGE` with a similar sampling strategy and we would surely love to include the discussions on them in the subsequent version.

**Reviewer 3 and 4** We thank both reviewers for the suggestions. Please refer to the additional result on the Amazon dataset at the top of this page (**Large dataset**). We would also like to further evaluate LAZYGCN on newly released datasets (OGB) and other settings in the subsequent version.

[Meta-Review · NeurIPS 2020]

All reviewers agree that this is a good contribution in speeding up the training of GCN. My recommendation is to accept. Please take the reviewers' comments into account in preparing the final version of the paper.